# Comparative Transcriptome Analysis of Two Types of Rye Under Low-Temperature Stress

**DOI:** 10.3390/cimb47030171

**Published:** 2025-03-03

**Authors:** Haonan Li, Jiahuan Zhao, Weiyong Zhang, Ting He, Dexu Meng, Yue Lu, Shuge Zhou, Xiaoping Wang, Haibin Zhao

**Affiliations:** 1Key Laboratory of Molecular Cell Genetics and Genetic Breeding in Heilongjiang Province, College of Life Science and Technology, Harbin Normal University, Harbin 150025, China; ryan_lhn@163.com (H.L.); zhaojh001201@163.com (J.Z.); 13359618736@163.com (W.Z.); ht15772790717@163.com (T.H.); 18645758939@163.com (D.M.); 18813192260@163.com (Y.L.); z1742900441@163.com (S.Z.); 2Pratacultural Research Institute, Heilongjiang Academy of Agricultural Sciences, Harbin 150086, China

**Keywords:** rye, low-temperature stress, transcriptome, cold-resistant candidate genes

## Abstract

Wheat is a crucial food crop, and low-temperature stress can severely disrupt its growth and development, ultimately leading to a substantial reduction in wheat yield. Understanding the cold-resistant genes of wheat and their action pathways is essential for revealing the cold-resistance mechanism of wheat, enhancing its yield and quality in low-temperature environments, and ensuring global food security. Rye (*Secale cereale* L.), on the other hand, has excellent cold resistance in comparison to some other crops. By studying the differential responses of different rye varieties to low-temperature stress at the transcriptome level, we aim to identify key genes and regulatory mechanisms related to cold tolerance. This knowledge can not only deepen our understanding of the molecular basis of rye’s cold resistance but also provide valuable insights for improving the cold tolerance of other crops through genetic breeding strategies. In this study, young leaves of two rye varieties, namely “winter” rye and “victory” rye, were used as experimental materials. Leaf samples of both types were treated at 4 °C for 0, 6, 24, and 72 h and then underwent RNA-sequencing. A total of 144,371 Unigenes were reconstituted. The Unigenes annotated in the NR, GO, KEGG, and KOG databases accounted for 79.39%, 55.98%, 59.90%, and 56.28%, respectively. A total of 3013 Unigenes were annotated as transcription factors (TFs), mainly belonging to the MYB family and the bHLH family. A total of 122,065 differentially expressed genes (DEGs) were identified and annotated in the GO pathways and KEGG pathways. For DEG analysis, 0 h 4 °C treated samples were controls. With strict criteria (*p* < 0.05, fold-change > 2 or <0.5, |log_2_(fold-change)| > 1), 122,065 DEGs were identified and annotated in GO and KEGG pathways. Among them, the “Chloroplast thylakoid membrane” and “Chloroplast” pathways were enriched in both the “winter” rye and “victory” rye groups treated with low temperatures, but the degrees of significance were different. Compared with “victory” rye, “winter” rye has more annotated pathways such as the “hydrogen catabolic process”. Although the presence of more pathways does not directly prove a more extensive cold-resistant mechanism, these pathways are likely associated with cold tolerance. Our subsequent analysis of gene expression patterns within these pathways, as well as their relationships with known cold-resistance-related genes, suggests that they play important roles in “winter” rye’s response to low-temperature stress. For example, genes in the “hydrogen catabolic process” pathway may be involved in regulating cellular redox balance, which is crucial for maintaining cell function under cold stress.

## 1. Introduction

Rye (*Secale cereale* L.) is an allopolyploid cereal crop. It has high nutritional value and excellent stress resistance. Consequently, it is prevalently employed in the global production of food, feed, and bioenergy. Moreover, in the marginal soil environment of the cold temperate zone in Northern Europe, rye is an extremely important food crop [1]. As the male parent donor of triticale, rye has played a role in improving important agronomic traits of wheat and thus has expanded the genetic diversity of wheat [2]. Rye has excellent resistance to various abiotic stresses, such as drought and salinity. However, there are differences in cold resistance among different rye varieties. For example, Weining rye exhibits relatively weak cold resistance. Through performing a comprehensive genome-wide comparative analysis of all transcription factors within plant species, valuable stress-resistant genes can be effectively identified [3]. In this study, the size of the recombinant genome sequence was 7.74 GB, among which the repetitive sequences accounted for 90.31%. The study thoroughly analyzed the gene duplication status within the wheat genome, explored the effects of these duplications on the starch biosynthesis genes in wheat, and investigated the gene expression characteristics related to the early heading traits. It also speculated on the chromosome regions and loci closely related to wheat domestication, which provided a crucial reference basis for mining the key genes in rye. The freezing damage induced by low-temperature stress constitutes one of the crucial elements that have an impact on the growth, development, and productivity of crops.

When the temperature undergoes a sharp decline, the entire life cycle of crops will be influenced. This might result in a substantial decrease in yield or even a grave circumstance of a complete crop failure [4]. Therefore, the genes in the pathways of winter rye can be identified as cold-resistant candidate genes. These candidate genes are of great reference value for exploring the cold-resistant mechanisms and mining cold-resistant genes in crops [5]. Currently, the cold response pathway regulated by DREB/CBF is widely recognized as one of the main ways for crops to resist low-temperature stress. Among them, DREB1/CBF proteins, as transcription factors, are involved in the expression of genes responding to cold stress in *Arabidopsis thaliana* [6]. In ginseng, the members of the NAC gene family related to cold stress include PgNAC05-2, PgNAC41-2, PgNAC48, PgNAC56-1, and PgNAC59 [7]. Furthermore, transcription factors like bZIP, WRKY, and bHLH also have a close association with low-temperature stress [8,9,10]. Low-temperature stress affects the fluidity of plant plasma membranes, photosynthesis, and metabolic processes through multiple signal transduction pathways. Moreover, the reaction of plants to cold stress encompasses a series of diverse transcriptional cascades and signal transduction procedures, which subsequently give rise to alterations in gene activity. Therefore, in-depth exploration of the functions of plant-related transcription factors can enhance the cold resistance of plants through precise regulation of gene activity. This will not only help to deepen the understanding of the plant’s stress response mechanism but also provide theoretical basis and technical support for improving the stress resistance of plants [11]. Although this section emphasizes the role of transcription factors in plants’ responses to low-temperature stress, this study encompasses other aspects as well. Low-temperature stress impacts various physiological processes in crops, including their entire life cycle, plasma membrane fluidity, photosynthesis, and metabolic pathways.

RNA-seq is a kind of high-throughput transcriptional dynamics analysis method [12]. This technology aims to analyze the relative expression levels of transcripts under different tissue backgrounds and has been applied as an important research tool in a variety of eukaryotes. The transcriptional expression profiles of rice [13], barley [14], and wheat when encountering different non-biological stresses have been thoroughly investigated [15]. In light of the significance of rye in agriculture and the impact of low-temperature stress on its growth, this study was designed to address several key questions. First, we aimed to comprehensively identify the differentially expressed genes between “winter” rye and “victory rye” under low-temperature stress. Understanding these differences is crucial for uncovering the genetic basis of cold-resistance variations among rye varieties. Second, we sought to elucidate the functional mechanisms of these differentially expressed genes. By exploring how these genes are involved in cold-resistance pathways, we can gain deeper insights into the complex regulatory networks that enable rye to adapt to cold environments. These investigations will not only contribute to a better understanding of rye’s cold-resistance mechanism at the molecular level but also provide a theoretical foundation for breeding cold-resistant rye varieties in the future.

## 2. Materials and Methods

### 2.1. Plant Materials and Low-Temperature Stress Treatment

For six varieties of rye *(Secale cereale* L.), including three winter-type rye varieties (“winter”, Hzhm8, Hzhm3) and three spring-type rye varieties (“victory”, 429,430), the rye seeds were sourced from the germplasm resource banks of the Grassland Research Institute of Heilongjiang Academy of Agricultural Sciences or the Key Laboratory of Molecular Cytogenetics and Genetic Breeding in Heilongjiang Province. Twenty seeds of each rye variety were cultivated in a greenhouse maintained at 25 °C under a 14 h light and 10 h dark cycle until reaching the three-leaf stage, after which they were relocated to an incubator set at 4 °C. “Winter” and “victory” were respectively exposed to gradient stress treatments for 0 h, 6 h, 12 h, 24 h, 48 h, 72 h, 96 h, 120 h, 144 h, and 168 h. Among them, the stress treatments at 0 h, 6 h, 24 h, and 72 h were used for transcriptome sequencing and phenotypic observation, while the other varieties and time periods were used for quantitative analysis. Three biological replicate experiments were carried out for each group.

### 2.2. Transcriptome Sequencing and Gene Information Prediction

First of all, in this study, the leaf tissues of rye were used. The total RNA was isolated with the Trizol kit from Invitrogen (Carlsbad, CA, USA), and the enrichment of mRNA was attained through the process of rRNA removal. Afterwards, the reverse transcription reaction was performed by using the PrimeScript RT MasterMix (Perfect Real Time) reverse transcription kit manufactured by TaKaRa (Kusatsu, Shiga, Japan). The transcriptome sequencing work was completed on the HiSeq2000 sequencing platform of Illumina (San Diego, CA, USA), and a total of 24 samples were sequenced.

During the data-processing stage, to ensure the reliability of the data, we used the Trimmomatic software (v0.36) to conduct strict filtering on the raw data. The contaminated combined reads (refer to the sequencing sequence fragments that are mixed in during the high-throughput sequencing process due to various reasons and will interfere with the accuracy and reliability of the experimental results), reads with the content of unknown bases (N) exceeding 5%, and low-quality reads (that is, the parts where the base quality value was lower than 15 and accounted for more than 20% of the total number of bases in the read) were removed. The reliable data (the data that can be used for subsequent accurate analysis after a series of strict screening and processing) obtained after filtering were assembled with the help of the Trinity software (version V2.4.0, Arlington, TX, USA). Subsequently, the Tgicl tool was utilized to conduct clustering processing on the transcripts and remove redundancies, and then Unigenes (a collection of non-redundant gene sequences obtained after a series of data processing in transcriptome research) were obtained. To further ensure the uniqueness of the Unigenes, Tgicl was used again to perform the redundancy removal operation on the Unigenes of the 24 samples. Eventually, the All-Unigenes set was obtained for subsequent in-depth analysis.

Regarding the assessment of the quality of the assembled transcripts, the single-copy orthologous database BUSCO was employed for alignment, and the integrity and precision of the transcriptome assembly were authenticated by contrasting with conserved genes. In addition, the Transdecoder tool was used to identify the candidate coding regions within the Unigenes. First, the longest open reading frames were searched for and determined. Then, in combination with the Blast comparison results against the SwissProt database and the search for Pfam protein homologous sequences using Hmmscan, the locations of the coding regions were ultimately predicted.

### 2.3. Sugar Content Determination and Injury Degree Assessment

In this study, to deeply explore the physiological responses of winter rye and spring wheat under different low-temperature stress conditions, we accurately determined the sugar content of plant samples and established a comprehensive and detailed assessment system for injury degree based on the quantification of visual damage indicators.

#### 2.3.1. Sugar Content Determination

The anthrone method, which is widely used in the analysis of plant sugar content, was employed to determine the sugar content in the samples. Before the experiment, leaves with consistent growth conditions and no obvious pests and diseases were selected from carefully cultivated winter rye and spring wheat plants.

Sample Preparation: 0.2 g of leaf samples were weighed and placed in a clean mortar. Then, 6 mL of phosphate-buffered saline with a pH value of 7 was added, and the mixture was thoroughly ground. The homogenate was carefully transferred to a centrifuge tube and centrifuged at 10,000 r/min for 10 min. The supernatant was collected for further use.

Reaction Process: 2 mL of the supernatant was taken into a test tube, and anthrone reagent was quickly added. The mixture was immediately shaken well to ensure full mixing. The test tube was first placed in an ice bath for 5 min to prevent overly vigorous reactions. Subsequently, it was placed in a constant-temperature water bath at 80 °C and heated for 10 min. After the reaction, the test tube was removed and cooled to room temperature. The absorbance was measured at a wavelength of 620 nm using a spectrophotometer.

Calculation of Results: Before the experiment, a series of glucose standard solutions with different concentrations (such as 0.1 mg/mL, 0.2 mg/mL, 0.3 mg/mL, 0.4 mg/mL, and 0.5 mg/mL) were prepared. The absorbance of each standard solution was measured according to the same procedures as above. The sugar content in the samples was calculated based on the standard curve established from the absorbance values of the standard solutions.

#### 2.3.2. Assessment of Injury Degree Based on the Quantification of Visual Damage Indicators

To accurately assess the injury degree of “winter” rye and “victory” rye under different experimental conditions, a comprehensive and detailed assessment system based on the quantification of visual damage indicators was constructed.

Quantification of Necrotic Spot Area Proportion: The proportion of the necrotic spot area to the total leaf area was carefully classified into different levels to measure the injury degree. A range of 0–5% is considered slight injury, denoted as level 1, and at this stage, only sporadic and extremely small necrotic spots appear on the leaves; 6–20% is mild injury, denoted as level 2, and necrotic spots begin to appear in local areas of the leaves, with an expanded area; 21–40% is moderate injury, denoted as level 3, and necrotic spots are more widely distributed on the leaves, significantly affecting the leaf appearance; 41–60% is severe injury, denoted as level 4, and a large area of the leaves is covered by necrotic spots, seriously threatening the growth of the plant; more than 61% is extremely severe injury, denoted as level 5, and the leaves are almost completely necrotic, and the plant is close to death.

Grading of Leaf-Curling Degree: The degree of leaf curling was used to grade the leaves. Slight curling, that is, the leaf edges curl slightly inward, is denoted as level 1, and the basic shape of the leaf has not changed significantly at this time; moderate curling, with the leaf-curling degree reaching 1/3–1/2 of the leaf width, is denoted as level 2, and the leaf shows a more obvious curling state; severe curling, with the leaf-curling degree exceeding 1/2 of the leaf width, is denoted as level 3, and the leaf is severely deformed, having a greater impact on photosynthesis and gas exchange.

Assessment of Leaf Color Change: A standard color chart was introduced to quantitatively assess the change in leaf color. The normal leaf color corresponds to a specific color number on the color chart. As the low-temperature stress intensifies, the leaves begin to turn yellow, red, etc. When the leaf color turns yellow, according to the yellow gradient series on the color chart, it is divided into light yellow (denoted as level 1), medium yellow (denoted as level 2), and dark yellow (denoted as level 3); if the leaves turn red, according to the red gradient series on the color chart, it is divided into light red (denoted as level 1), medium red (denoted as level 2), and dark red (denoted as level 3). The more obvious the color change, the more severe the damage to the plant.

Comprehensive Assessment of Multiple Indicators: To more comprehensively and accurately assess the injury degree of plants, the levels of necrotic spot area proportion, leaf-curling degree, and leaf color change were comprehensively considered. For example, a weighted-average method was used to calculate the comprehensive injury index. A higher weight (such as 0.5) was assigned to the level of necrotic spot area proportion, the weight of the leaf-curling degree level was 0.3, and the weight of the leaf color change level was 0.2. The comprehensive injury index was calculated through the formula “Comprehensive Injury Index = Level of Necrotic Spot Area Proportion × 0.5 + Level of Leaf Curling Degree × 0.3 + Level of Leaf Color Change × 0.2”. The higher the index, the more severe the injury degree of the plant.

### 2.4. Gene Annotation

The assembled Unigenes were annotated with the utilization of seven functional databases, namely KEGG (Kyoto Encyclopedia of Genes and Genomes), GO (Gene Ontology), NR (Non-redundant database), NT (Non-redundant Nucleotide Database), SwissProt, Pfam, and KOG (Clusters of Orthologous Groups of protein database for eukaryotes). The functions of Unigenes were determined in accordance with the characteristics of the transcription factor families depicted in PlantTFDB. For the sake of annotation, the DIAMOND software (v2.1.8) was utilized to conduct a comparison between the differentially expressed genes and PRGdb (Table A1).

### 2.5. Differentially Expressed Genes (DEGs) and Analysis of Functional Enrichment

For statistical testing, the DESeq2 software (v1.40.2) package is employed to conduct 16 sets of tests for differentially expressed genes. DESeq2 is a commonly used tool for analyzing differential gene expression in RNA-seq data. It evaluates whether the changes in gene expression are statistically significant through statistical analysis of the gene expression levels among samples.

Regarding the threshold setting, when performing functional enrichment analysis, the phyper function in the R software (v4.3.1) is used for the enrichment analysis, and the *p*-values are corrected by the FDR (False Discovery Rate). Generally, genes with a Q-value (corrected *p*-value) ≤ 0.05 are considered as genes with significantly differential expression among different sample groups. The setting of this threshold is a crucial criterion for determining whether a gene is a differentially expressed gene. Genes with a value lower than this threshold indicate that their expression changes are statistically significant, that is, they are identified as differentially expressed genes.

First, Bowtie2 was employed to map the clean reads to the genomic sequence. Subsequently, RSEM was utilized to compute the gene expression levels of each sample. Then, DESeq2 was applied to perform 16 groups of DEGs tests following the approach described. The materials utilized are presented (Table 1).

Based on the results of DEGs, we classified the differential genes into functional categories by using the GO and KEGG annotation results and official classifications (the gene function classification standards and systems that are widely recognized and commonly used by authoritative institutions or within the academic field). The enrichment analysis was carried out using the phyper function in R software, and then the *p*-values were corrected by FDR (False Discovery Rate). Generally, functions with a Q-value ≤ 0.05 were regarded as significantly enriched.

### 2.6. qRT-PCR Identification of Low-Temperature-Tolerance-Related Genes in Winter Rye

To screen for the genes associated with low-temperature tolerance in “winter”, the young leaves of three winter rye varieties (“winter”, Hzhm8, Hzhm3) and three spring rye varieties (“victory”, 429,430) were selected as experimental materials in this study and qRT-PCR assays were conducted under the gradients of 0 h, 12 h, 24 h, 48 h, 72 h, 96 h, 120 h, 144 h, and 168 h, respectively. Initially, 450 ng of total RNA was extracted for the reverse transcription process.

We selected β-tubulin as our housekeeping gene. It is stably expressed in different tissues of wheat and under various experimental conditions. We have also verified their expression stability in our own preliminary experiments. For example, the expression stability of several candidate housekeeping genes was analyzed using the geNorm and NormFinder algorithms, and β-tubulin showed the highest stability score.

The fold change was calculated using the 2^−ΔΔCt^ method. First, we calculated the ΔCt value for each target gene, which is the difference between the Ct value of the target gene and the Ct value of the housekeeping gene in the same sample (ΔCt = Ct target gene − Ct housekeeping gene). Then, we calculated the ΔΔCt value by subtracting the average ΔCt value of the control group from the ΔCt value of each experimental group (ΔΔCt = ΔCt experimental group − average ΔCt control group). Finally, the fold change was calculated as 2^−ΔΔCt.^

Differential expression level: We first screened according to the fold change and *p*-value of the differentially expressed genes (DEGs). Genes with a fold change greater than 2 (up-regulated or down-regulated) and a *p*-value less than 0.05 were considered to be significantly differentially expressed genes. This step helped us focus on genes with significant changes in expression levels under different low-temperature stress conditions. Biological function relevance: We gave priority to genes that are likely to be involved in important biological processes related to the low-temperature stress response, such as cold acclimation, osmotic regulation, and antioxidant defense. We referred to the Gene Ontology (GO) annotations and Kyoto Encyclopedia of Genes and Genomes (KEGG) pathway analysis results to identify genes with relevant functions.

Representativeness of different expression patterns: We aimed to select genes that represent different expression patterns among the DEGs, including genes that are continuously up-regulated, continuously down-regulated, or show transient changes in expression during low-temperature treatment.

Primer3 was used to design primers for selected gene sequences (Table A3). The PCR reaction system was set to 20 μL, which contained 12.5 μL of (SybrGreen qPCR Master Mix (2×)), 0.5 μL each of the upstream and downstream primers, 10.5 μL of ddH_2_O, and 1 μL of cDNA. The amplification cycle parameters were set as follows: pre-denaturation at 95 °C for 10 min, followed by 35 cycles of denaturation at 95 °C for 15 s, annealing at 60 °C for 1 min, and then additional steps of denaturation at 95 °C for 15 s, annealing at 60 °C for 1 min, denaturation at 95 °C for 15 s, and a final annealing at 60 °C for 15 s. For each gene, three biological replicate experiments were carried out in these six varieties of rye based on the treatment under the same low-temperature condition for different durations. Meanwhile, the housekeeping gene was used as the corresponding control, and finally, the gene expression level data were obtained (Table A2).

## 3. Results

### 3.1. Growth Alterations of Two Varieties of Rye Under Low-Temperature Stress

In this research, two rye varieties, specifically “winter” and “victory”, were subjected to a treatment at 4 °C for 6 h, 24 h, and 72 h, respectively, after which their growth states were contrasted. Previous studies have shown that an increase in soluble sugars can enhance the plant’s ability to tolerate low temperatures by regulating the osmotic potential and protecting cellular components. Measuring the soluble sugar content of the “winter” rye and “victory” rye varieties under cold stress enables us to compare their physiological responses and potentially identify differences in their cold tolerance mechanisms.

The results showed that both varieties of rye exhibited lodging and wilting phenomena after low-temperature treatment. As the stress time continued to extend to 72 h, at 6 h of cold stress, the leaf margins of “victory” rye were slightly curled, while the leaves of “winter” rye were relatively flat. By 72 h, the leaves of “victory” rye turned light green and wilted severely, whereas the leaves of “winter” rye, although also showing signs of stress, remained a darker green and the wilting was less obvious. (Figure 1a). In terms of the changes in their physical and chemical properties, for the soluble sugar content (Figure 1b), at 72 h, the soluble sugar content in “winter” was greater than that in “victory”, and a notable disparity in sugar content values was observed between the two varieties.

### 3.2. De Novo Assembly and Coding Sequence Identification and Analysis of the Rye Transcriptome

In this study, transcriptome sequencing was performed on 24 samples, generating a total of 1061.8 M reads. After data filtering, 1016.97 M high-quality data were finally obtained (Table 2), approximately 152.41 GB, among which the Q20 quality score was greater than 96.47% and the Q30 was greater than 91.2%. Subsequently, the Tgicl software (v2.1) was used to conduct clustering and redundancy removal processing on these transcripts, and finally, 144,371 Unigenes were obtained, with a total length of 222,447,790 bp, an N50 of 2178 bp, an N90 of 850 bp, and a GC content of 49.38%.

First of all, the longest open reading frames (ORFs) were retrieved to pinpoint the candidate coding regions inside the Unigenes by means of the TransDecoder software (v5.4.0). Then, these ORF sequences were put through Blast alignment with the SwissProt database, and the Hmmscan software (v3.2.1) was used to search for the homologous sequences of the Pfam protein family. Eventually, 94,449 coding regions (CDS) were predicted, with a total length of 93,447,723 nt. The maximum length of the coding regions was 14,682 nt, the minimum length was 297 nt, and the GC content was 53.76%. Most of the CDS lengths were concentrated between 300 nt and 700 nt (Table 3).

### 3.3. Gene Annotation, Transcription Factor Identification, and Functional Classification Analysis of the Cold-Resistant Gene Domains in Rye

In this research, following the removal of redundancy by Tgicl, a sum of 144,371 Unigenes were annotated (Table 4). Out of these, 114,619 Unigenes were annotated within the NR database. Afterwards, these Unigenes were further annotated with respect to the GO database, yielding 80,822 GO-annotated Unigenes. The annotation results of the NR database indicated that most of the matched homologous species were plants of the Poaceae family, among which the matching degree with wheat subspecies was the highest. Specifically, the matching degrees of *Triticum turgidum* subsp. *durum* and *Aegilops tauschii* subsp. *tauschii* were 40.5% and 25.64%, respectively.

In the KEGG database, 59.90% of the Unigenes were annotated. Among them, 69,358 Unigenes were annotated by both the KEGG and GO databases simultaneously (Figure 2). The number of Unigenes annotated by all databases was 49,045, whereas the quantity of Unigenes that remained unannotated by any database was 15,185.

The GO database provided a total of 80,822 annotations for the physiological procedures, molecular capabilities, and cellular constituents of the Unigenes, among which there were 28 pathways enriched by GO. The category with the largest number of annotations was “cellular anatomical entity” (65,334 annotations), followed by “binding” (57,305 annotations), which belonged to the annotation categories of cellular component and molecular function, respectively. Within the KEGG database, a cumulative total of 19 branches were annotated, with 11 of those branches being associated with metabolism.

The pathway with the largest number of annotated Unigenes was the “Metabolic pathways”, with a total of 52,197, accounting for 60.36% of all KEGG annotations. The only pathway related to the cellular component process was the “Transport and catabolism” pathway.

In the KOG database, the Unigene sequences were used for the classification of gene homologs. The classification information included “Nuclear Structure” (271), “RNA processing and modification” (3862), and “Signal transduction mechanisms” (13,129) (Figure 3a,b). In addition, the prediction and classification of transcription factor (TF)-encoding genes indicated that 3013 Unigenes were annotated as transcription-factor-encoding genes. Among them, the MYB family (306) and the bHLH family (248) had the largest number of genes. Moreover, shared TF family genes linked to abiotic stress, like NAC, WRKY, and bZIP, were additionally annotated (Figure 3c).

The DIAMOND software (v2.1.8) was used to align the genes with the PRGdb database. The results showed that 12,008 Unigenes were annotated as genes related to plant disease-resistant domains, which were divided into 12 main domain types in total. Among them, the majority were 2–3 composite domains, for example, RLP (receptor serine-threonine kinase-like domain/extracellular leucine-rich repeat) and TNL (Toll-interleukin receptor-like domain/nucleotide binding site/leucine-rich repeat) (Figure 3c).

### 3.4. Analysis of DEGs Between “Winter” and “Victory” Under Low-Temperature Stress Reveals Their Gene Expression Characteristics

A total of 122,065 DEGs were detected among the 16 groups based on the analysis of the gene expression levels of each sample. Among these, 61,158 genes were up-regulated and 60,907 genes were down-regulated. The analysis showed that the number of DEGs in the “victory” group was higher than that in the “winter” group (Figure 4a,b). Further comparison of the number of DEGs between the “winter” group and the “victory” group revealed that as the stress time extended, the number of DEGs gradually increased, yet this increasing trend was relatively gentle. In the four groups of experiments comparing “winter” and “victory” under cold stress at 0 h, 6 h, 24 h, and 72 h, a total of 34 genes showed differential expression in each group, whereas merely 7247 distinct differentially expressed genes (DEGs) were identified in the two rye control groups (Figure 4c,d). These outcomes suggest that the disparities between the “victory” group and the “winter” group in stress responses exhibit notable gene expression characteristics and some key genes show a common response in all groups.

### 3.5. GO and KEGG Enrichment Analyses of Differentially Expressed Genes in “Winter” and “Victory” Ryes Under Low-Temperature Stress Reveal Their Cold Tolerance Mechanisms

After annotating the DEGs with GO and KEGG pathways, pathways related to cold stress were significantly enriched, which is of great significance for understanding the differences in the metabolic processes between “winter” and “victory” and for the discovery of their cold-tolerant candidate genes. We conducted GO enrichment analyses on the six groups of DEGs at three levels, specifically cellular component (CC), biological process (BP), and molecular function (MF) (as shown in Figure 5a).

At the level of CC (cellular component), 25 GO terms were enriched in “winter”, while 28 GO terms were enriched in “victory”. Specific pathways such as “plastid-encoded Plastid RNA polymerase complex” and “plastid chromosome” were significantly up-regulated or down-regulated in the six groups of treatments of winter (as shown in Figure 5b), especially in the groups of D-CK vs. D-72 h and D-24 h vs. D-72 h. In addition, although both rye varieties were enriched in the “chloroplast thylakoid membrane” pathway at the CC level, the degrees of enrichment were different. In “victory”, significant differences were only observed between S-6 h and S-72 h (as shown in Figure 5c), while in the six groups of treatments of “winter”, significant up-regulation or down-regulation was shown. This indicates that although chloroplast formation genes are related to the cold-resistance pathway, they may be unrelated to the rye varieties.

In the BP (biological process) pathways, there were 24 more GO-enriched pathways in “winter” than in “victory”. In total, there were 29 commonly enriched pathways, including “polysaccharide metabolic process”, ”photorespiration”, and “glyoxylate cycle”, etc. These pathways imply that the cold tolerance of rye might be attained by modulating the decomposition of polysaccharides and alterations in photorespiration. It is worth noting that “winter” showed an enrichment of the “hydrogen peroxide catabolic process” under low-temperature stress, while such an enrichment was not observed in “victory”. This might be due to the decreased activity of catalase in “victory” under low temperatures.

In the MF (molecular function) pathways, both rye varieties were enriched in “Monooxygenase activity”, but the difference was reflected in the different amounts of up-regulated DEGs at different time periods. For example, in “winter”, the up-regulated amounts of DEGs significantly increased when comparing D-CK with D-24 h and D-CK with D-72 h, while in “victory”, an increase in the up-regulated amounts was shown at S-6 h and S-24 h. These results indicate that there is an asynchronous phenomenon in the responses of the two rye varieties in terms of “Monooxygenase activity”.

In the KEGG pathway analysis, a total of 38 pathways in “winter” and 16 pathways in “victory” were annotated. Pathways with a Q value ≤ 0.05 were considered to be significantly enriched. Although the annotations of the metabolic pathways of the two rye varieties were mostly similar, such as “tryptophan metabolism”, ”histidine metabolism”, and “glyoxylate and dicarboxylate metabolism”, The GO enrichment outcomes suggested that the cold tolerance mechanism of rye could be highly associated with the glyoxylate cycle.

Overall, the number of GO and KEGG pathway enrichments in “winter” was significantly higher than that in “victory”. Especially after 72 h of stress, the accumulation and up-regulation of DEGs in “winter” were more remarkable. This indicates that “winter” has a broader cold-resistance mechanism and shows stronger adaptability under long-term low-temperature stress.

### 3.6. Utilizing the qRT-PCR Technique to Select Genes Relevant to Cold Tolerance in Winter Rye

We subjected six varieties of rye to a treatment at −4 °C, and observed and compared the changes in leaf injury levels at different times (Figure 6).

To conduct a more in-depth exploration of the response of rye to low-temperature stress, we utilized the qRT-PCR technique to analyze the expression changes of eight cold-tolerant candidate genes (Figure 7): ScRVE1 (it is a gene related to transcription factors, and its specific function may involve regulating the gene transcription process.), ScCDC5 (it belongs to the genes related to transcription factors; transcription factors may play a crucial role in biological processes such as the regulation of the cell cycle.), ScPME18 (It is also a gene related to transcription factors; it is speculated that it exerts its function by regulating gene transcription in the physiological processes of plant cells.), ScWRKY55 (as a gene related to transcription factors, the WRKY family of transcription factors plays an important role in various physiological processes of plants, such as responding to biotic and abiotic stresses.), ScHsP (this is a gene related to proteins, regarding the specific function of the protein encoded by this gene.), ScAAE7 (it belongs to the genes related to proteins.), ScHs16 (it is also a gene related to proteins.), ScMYB92 (this gene is a differentially expressed gene related to metabolism.) MYB family genes usually play important roles in the metabolic regulation, growth and development, and other processes of plants. The results demonstrate that the expressions of the ScMYB92, ScCDC5, and ScAAE7 genes were significantly up-regulated in winter rye varieties. In particular, after the “winter” rye was treated for 168 h, the transcriptional abundance of theScAAE7 gene reached the highest.

In contrast, in the response of the spring rye variety 430 to low-temperature stress, the expression levels of the *ScMYB92* and *ScAAE7* genes were significantly down-regulated. It is remarkable that the transcriptional abundance of the *ScCDC5* gene was notably up-regulated in winter rye yet markedly down-regulated in spring rye, showing a good ability to distinguish between varieties. The *ScPME18* gene was only up-regulated in “winter” rye, demonstrating good specificity for cold-tolerant genes. These results indicate that the selected candidate genes have the potential in distinguishing the low-temperature responses of different rye varieties. Especially, the *ScCDC5* and *ScPME18* genes might have a vital part to play in the forthcoming research concerning cold-resistance mechanisms and rye breeding.

## 4. Discussion

The transcriptomes of rye varieties “winter” and “victory” under different conditions were analyzed in this study. By comparing the changes in low-temperature stress at 0 h, 6 h, 24 h, and 72 h in an environment of 4 °C between the two varieties, as the length of low-temperature stress grew, the distinctions in the transcriptome data of “winter” became more pronounced. With the help of gene annotation, transcription factor identification, and other methods, genes and signal transduction pathways related to low-temperature stress were successfully discovered. This study’s results have laid a solid foundation for the subsequent exploration of rye’s genetic traits and molecular mechanisms. They are also of great significance in rye genetic breeding, providing a powerful theoretical basis and data support for it.

The differential pathways between “winter” and “victory” rye provide important clues for understanding the cold-resistance mechanism. While the mere existence of more pathways in “winter” rye does not guarantee cold resistance, it indicates a more complex and potentially more efficient response system. We hypothesize that the coordinated regulation of genes within these pathways, along with other factors such as gene expression levels and protein–protein interactions, contributes to “winter” rye’s enhanced cold tolerance. Future experiments, such as gene knockout and overexpression studies, are needed to directly verify the functions of these genes and pathways in cold resistance. Low-temperature stress can damage the plasma membrane environment of plants and then affect their normal physiological functions. When the temperature changes, the plasma membrane will transform from a semi-fluid state to a semi-crystalline state [16]. Meanwhile, cell wall polymers will also be widely modified by various cell wall hydrolases (i.e., glycosyl hydrolases), which will subsequently lead to significant changes in their biophysical properties. It is worth mentioning that in the rice genome, as many as 49 members of the PMEI family were discovered by Hong et al. [17]. Quantitative real-time PCR technology, in conjunction with molecular-level expression analysis, was utilized to perform a comprehensive examination of the transcriptional levels of these members. The outcomes indicated that the members of the PMEI family were finely regulated in both temporal and spatial dimensions and could respond to multiple stressors. Based on the above findings, we hypothesize that upon exposure to cold stress, by regulating the activities of plant cell wall-related hydrolases, *ScPMEI8* might have the capacity to diminish the harm inflicted by low-temperature stress on the plant cell wall, helping plants better adapt to the cold environment. We also found that, in addition, *ScAAE7* may be involved in activating the process of exogenous acetate entering the glyoxylate cycle; during lipid mobilization, this mechanism plays a crucial role in preventing peroxisomal carbon loss. Jay M et al.’s study on the acetate/glucose dioxygen growth of rice cell cultures further confirmed the importance of acetate as a respiratory metabolite [18]. They found that acetate could inhibit glucose uptake, an increase in the activity of the glyoxylate cycle enzyme isocitrate lyase accompanied the preferential use of acetate. To better cope with changes in the external environment, this mechanism may help regulate the transition from heterotrophy to autotrophy in developing seedlings.

Furthermore, transcription factors play a significant part in enhancing plant cold tolerance. By regulating the expression of downstream genes, they can initiate cascading reactions in relevant metabolic pathways, thereby enhancing plants’ adaptability to low temperatures [19]. That 3013 Unigenes were annotated as transcription factors (TFs) was shown by the results of this study. Some common families related to abiotic stress resistance, such as NAC [20], WRKY [21], and bZIP [22], were also annotated (Figure 3c). This was highly consistent with previous research results, indicating that transcription factors such as MYB and bHLH were closely related to cold tolerance.

Furthermore, previous studies have demonstrated that *MdMYB5523* and *MdMYB0970* were relatively sensitive to cold stress, and these two genes were positive regulators of plants’ responses to cold stress [23], playing important roles in cold tolerance. Suggesting that MYB might be involved in the regulatory process of plants’ responses to cold stress, MYB was up-regulated under long-term cold stress in our results. The transcription factors involved in the regulation of “winter”’s cold adaptation could be identified by comparing the transcription factors that were up-regulated or down-regulated under different conditions.

We speculate that amidst low-temperature stress conditions, there are notable disparities in the transcriptome gene expressions between the two varieties, “winter” and “victory”. Among them, *ScCDC5*, as an important component of the MAC complex, may regulate the plant’s defense responses via transcriptional regulatory pathways, and it is crucial for maintaining the plant’s innate immunity. Recently, both of these two types of RNAs, specifically microRNA (miRNA) and small interfering RNA (siRNA), have been verified to participate in plant growth and development as well as innate immune regulation. Palma et al.’s research further demonstrated that the MAC complex can promote the regulatory processes mediated by them [24]. In addition, *ScCDC5* is also involved in mRNA splicing and the control of the cell cycle, which further reflects its central position and important role during the course of plant growth and development.

On the other hand, there is an interaction between *ScRVE1* and the morning-phase transcription factors. This finding integrates the auxin signaling pathway and the biological clock. It regulates the level of free auxin in a circadian rhythm-specific manner by binding to the Evening Element (EE) in the promoter. Notably, *ScRVE1* has been recognized as a negative regulator of freezing tolerance, and the cold tolerance of plants may be closely related to its expression pattern.

Meanwhile, *ScWRKY55* combines with a common elicitor-responsive cis-acting element, the W-box, by virtue of its unique sequence specificity, and then regulates the activity of plant metabolic pathways. Such a regulatory mechanism may enable plants to respond more flexibly to environmental changes. In addition, Gibbs et al.’s research pointed out that in the process of Arabidopsis lateral root development, MYB93 is a newly discovered negative regulator, endowing plants with great plasticity to adapt to the continuously changing external environment. In this study, we discovered another MYB transcription factor, *ScMYB92*, whose expression level rose remarkably under cold stress circumstances. By regulating the structure and function of plant roots, we speculate that *ScMYB92* may enable rye to better withstand the challenges brought by the cold environment.

Therefore, we compared the transcriptome sequences of these two rye varieties. In other studies, some additional factors, encompassing the synthesis and accumulation of compatible solutes as well as the synthesis of cold acclimation-induced proteins, may also cause damage to plants due to cold stress [25]. Low temperature may also lead to severe cell dehydration [26]. When the intercellular solution freezes, it will also cause mechanical damage to cells. Growth retardation, wilting, and weak seedlings will be exhibited by plants, and ultimately, the crop yield and quality will be impacted [27].

Abscisic acid (ABA), being a vital plant hormone, plays a wide range of regulatory roles in the growth and development process of plants as well as in their responses to adverse conditions [28]. Specifically, ABA [29] is a key regulatory factor for plants to respond to abiotic stress and has a profound impact on the metabolic pathways of plants in a low-temperature environment. Through the analysis of GO data, we observed that as the duration of stress increased, the number of differentially expressed genes in metabolic pathways gradually decreased in both the “winter” and “victory” varieties. The suppressive impact of the low-temperature setting on the expression of these pathways was presumably the cause. Earlier investigations in Arabidopsis thaliana have disclosed a comparable occurrence [30]. Specifically, the expressions of the abscisic acid-responsive element promoters RD29A [31], COR15 [32], COR47 [33], RD22 [34], and P5CS were all significantly decreased or even completely blocked under low-temperature stress [35], which further hindered the normal progress of relevant metabolic pathways. This finding is highly consistent with our analysis results, further confirming the central role of ABA in the plant’s low-temperature adaptation mechanism.

Low-temperature stress has adverse effects on plant growth and crop yield. Some plants express a series of cold-responsive genes during the cold acclimation process [36] to mitigate the harm caused by low-temperature stress. At 6 h and 24 h, transcriptome analysis showed that genes related to biological processes, cell composition, and molecular functions were differentially expressed, indicating that cold stress has a significant impact on plants, which is consistent with the results of Rui Shi et al.; comparisons between other groups also showed the same results [37]. “Winter” can enhance cold resistance and resist cold damage by synthesizing extracellular components such as Keratin, Wood cork and wax properties, RNA transport, Phenylpropanoid biosynthesis, and Splicing body synthesis [38].

To verify the data of RNA-seq [39], we used qRT-PCR to detect eight differentially expressed genes [40]. *ScMYB92* is a differentially expressed gene associated with metabolism. *ScAAE7* is a differentially expressed gene linked to proteins, and *ScCDC5* is a differentially expressed gene associated with transcription factors. When plants are subjected to low-temperature stress, it is possible that the differential expression of the *ScMYB92* gene modulates plant metabolism, accelerates processes such as enzyme utilization and water synthesis, and thus regulates plant photosynthesis. Comparisons between other groups also showed the same results, which is consistent with the results of Rui Shi et al., indicating that cold stress has a significant impact on plants.

Therefore, *ScCDC5* can regulate the activity of the low-temperature transcriptome, elevate the expression of the plant’s response transcriptome to low temperature, and thus cope with the impairment brought about by low-temperature stress. Therefore, the fundamental reason why cold-sensitive plants possess a greater capacity to change temperature lies in the precise response to low temperature at the plant molecular level. Essentially, the attainment of plant cold acclimation and cold-tolerance capacity is the result of the expression regulation of a series of transcription factors and cold-resistance genes in plants under the stimulation of chilly temperature [41]. In this study, in terms of growth status, the degree of atrophy of the “victory” variety was significantly greater than that of the “winter” variety. This difference further confirms that different plant varieties have differences in adaptability and cold-resistance under low-temperature stress.

The relative expression tendencies of the eight genes in this article were highly in line with the RNA-sequencing outcomes signifying that RNA-seq analysis exhibits high fidelity and is suitable for further research undertakings. Beginning from the viewpoints of transcription factors that are highly relevant to plant cold tolerance and the production of associated proteins, this article mined candidate genes related to cold resistance. We discovered that within the differentially expressed genes associated with transcription factors, the quantity of such genes grew as the length of the stress period was prolonged. Taking the *ScWRKY55* gene as an example, at 6 h, 24 h, and 72 h, its expression levels were 4.98, 8.14, and 11.22, respectively, showing a significant upward trend.

In “winter”, after being subjected to low-temperature stress for 12 h, 24 h, and 72 h, within numerous GO terms associated with molecular functions, such as “sequence-specific DNA binding” and “DNA-binding transcription factor activity”, it enriched CK, and at the cellular function level, it was closely associated with the cell nucleus. In our study, *ScWRKY55* was annotated with 10 GO terms related to cells, including a related protein 37A in the vacuolar protein category. This gene was also annotated with nine GO terms, including signal transducer and activator of transcription 6. There were genes related to sequence-specific DNA binding and DNA-binding transcription factor activity, which were labeled as WRKY transcription factors, among the GO terms related to molecular functions. WRKY transcription factors are counter-regulatory proteins involved in plant immune responses, development, and senescence. Involved in plant growth and development, stress reactions, and the biosynthesis of secondary metabolites like terpenoids [42], the bZIP transcription factor represents a significant family in plants. Meanwhile, the candidate gene *ScWRKY55* obtained in this study was also annotated in the bZIP family, which was consistent with the qRT-PCR results. To sum up, all the eight candidate genes screened out were related to cold stress, laying a foundation for future rye genetic breeding and being valuable resources for rye cold-resistant breeding.

## 5. Conclusions

Through the transcriptome analysis of two rye varieties under low-temperature stress, cold-resistant candidate genes of rye varieties were screened out.

## Figures and Tables

**Figure 1 cimb-47-00171-f001:**
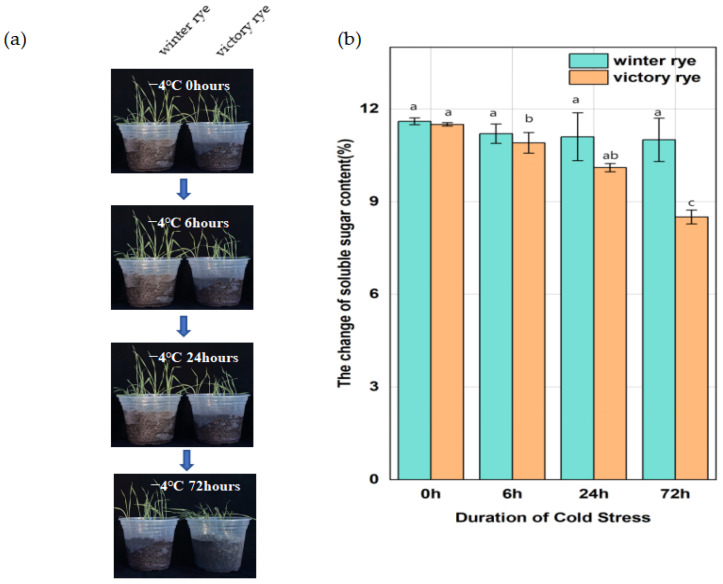
Changes in leaf growth status and soluble sugar content of “winter” rye and “victory” rye at different time points under different low-temperature stresses. (**a**) The left and right sides of each sampling time image depict the leaf growth conditions of “winter” rye and “victory” rye, respectively; (**b**) Bar chart presenting the variations in the soluble sugar content within the leaves of “winter” rye and “victory” rye following 0 h, 6 h, 24 h, and 72 h of low-temperature stress. Green symbolizes “winter” and orange stands for “victory”. The treatment at 0 h served as the control group, while the treatments at other time points were the experimental groups. The letters such as a, b, and c in the figure are the results marked based on statistical significance tests (such as multiple comparisons after analysis of variance). There is no significant difference in soluble sugar content among groups marked with the same letter; there are significant differences in soluble sugar content among groups marked with different letters.

**Figure 2 cimb-47-00171-f002:**
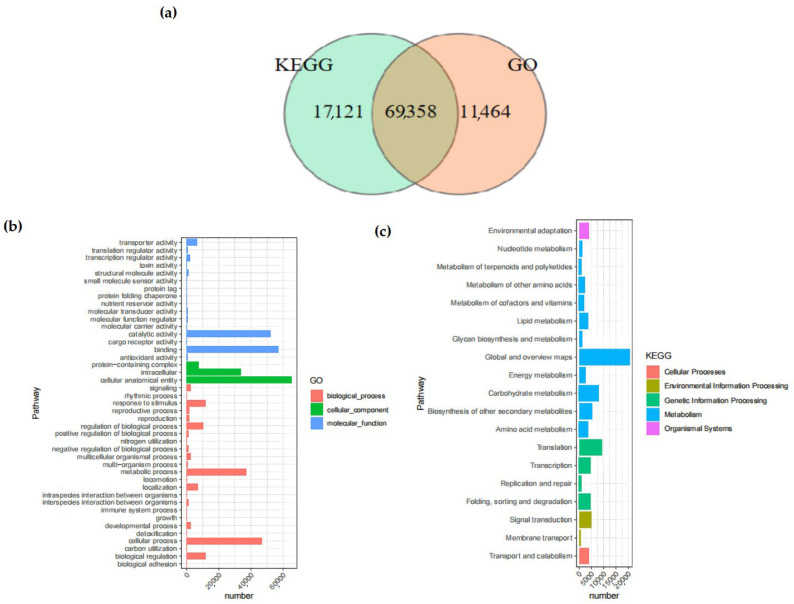
Venn diagrams of Unigenes. (**a**) Comparison between Unigenes in KEGG analysis and those in GO analysis. The total number of Unigenes in the comparison combination is represented by the sum of the numbers in each large circle. The overlapping circles represent the quantity of shared Unigenes in each combination; (**b**) GO enrichment distribution of Unigenes. It shows the enrichment of Unigenes in pathways among biological processes, molecular functions, and cellular components; (**c**) KEGG enrichment distribution of Unigenes. It shows the enrichment of Unigenes in pathways among cellular pathways, environment, genetic information, metabolism, and organic systems. The abscissa indicates the enrichment factor, while the ordinate represents the names of the pathways. The length of the horizontal lines represents the number of Unigenes in the pathways, and the colors of the dots correspond to different pathways.

**Figure 3 cimb-47-00171-f003:**
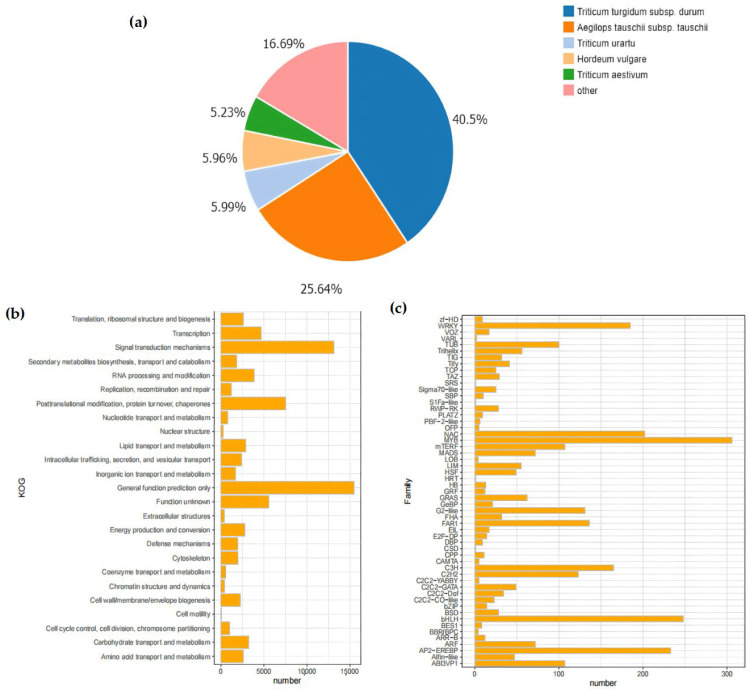
Analysis of database annotation results. (**a**) Pie chart of Unigenes annotated by the NR database, encompassing those in *Lolium*, *Aegilops tauschii*, *Triticum urartu*, *Hordeum vulgare*, *Triticum aestivum*, and so forth. The percentage of the pie chart indicates the quantity of Unigenes in each species; (**b**) The vertical axis lists different KOG functional categories, including translation, ribosome structure and biogenesis, transcription, signal transduction mechanisms, etc. The horizontal axis represents the number of Unigenes, ranging from 0 to 15,000. Each functional category corresponds to an orange bar, and the length of the bar indicates the number included in that functional category. From the figure, the quantitative distribution of different KOG functional categories can be intuitively seen; (**c**) Histogram of gene family distribution. In the figure, the horizontal axis represents the number of Unigenes distributed, ranging from 0 to 300, and the vertical axis lists different gene families. Each gene family corresponds to an orange bar, and the height of the bar indicates the quantity encompassed within that gene family.

**Figure 4 cimb-47-00171-f004:**
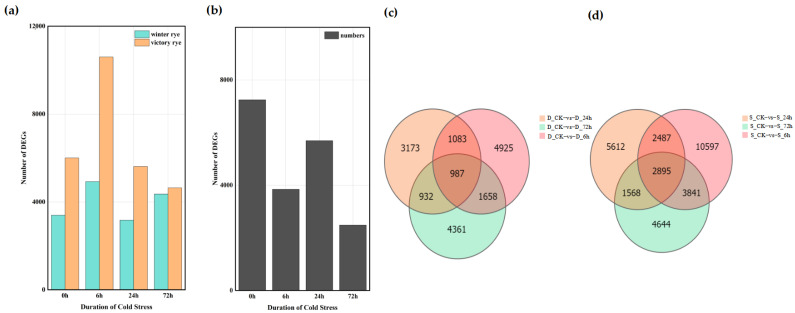
Distribution of differentially expressed genes in “winter” rye and “victory” rye at different time points after cold stress. (**a**) The number of differentially expressed genes in various samples. Green and yellow represent the differentially expressed genes of “winter” rye and “victory” rye varieties, respectively; (**b**) The quantity of differentially expressed genes in “winter” rye and “victory” rye at diverse time points under cold stress. The treatment at 0 h served as the control group, while the treatments at other time points were the experimental groups. (**c**) shows the comparison between DEGs at different treatment times in “winter” rye (D) and DEGs in the control group CK; (**d**) shows the comparison between DEGs at different treatment times in “victory” rye (S) and DEGs in the control group CK. The treatment at CK served as the control group, while the treatments at other time points were the experimental groups. The total number of DEGs in the comparison combination is represented by the sum of the numbers in each large circle. The number of common DEGs in each combination is represented by the overlapping circles.

**Figure 5 cimb-47-00171-f005:**
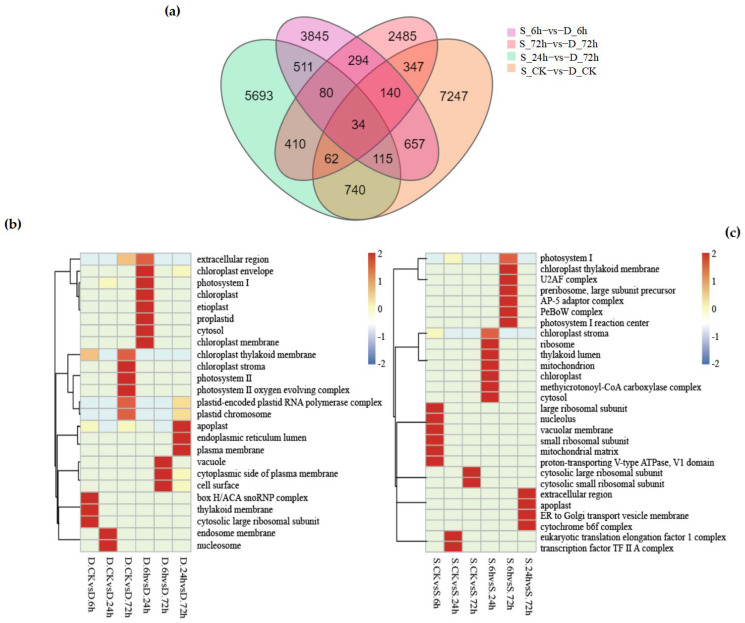
(**a**) displays the variations in differentially expressed genes (DEGs) between “Winter Rye” and “Victory Rye” at different treatment times: control (ck), 6 h, 24 h, and 72 h. The total count of numbers inside each large circle signifies the aggregate number of DEGs in the respective comparison combination, with overlapping circles indicating the shared number of DEGs among these combinations. (**b**) presents a comparison of various cellular component pathways in the “Winter Rye” variety under the same treatment conditions, specifically at control (ck), 6 h, 24 h, and 72 h. Here, the vertical axis denotes the pathway name, the horizontal axis represents the treatment time, and the color intensity indicates the enrichment level of DEGs in a given pathway. (**c**) shows the corresponding comparison for the “Victory Rye” variety. Note: In the figure, “Winter Rye” is abbreviated as D and “Victory Rye” as S.

**Figure 6 cimb-47-00171-f006:**
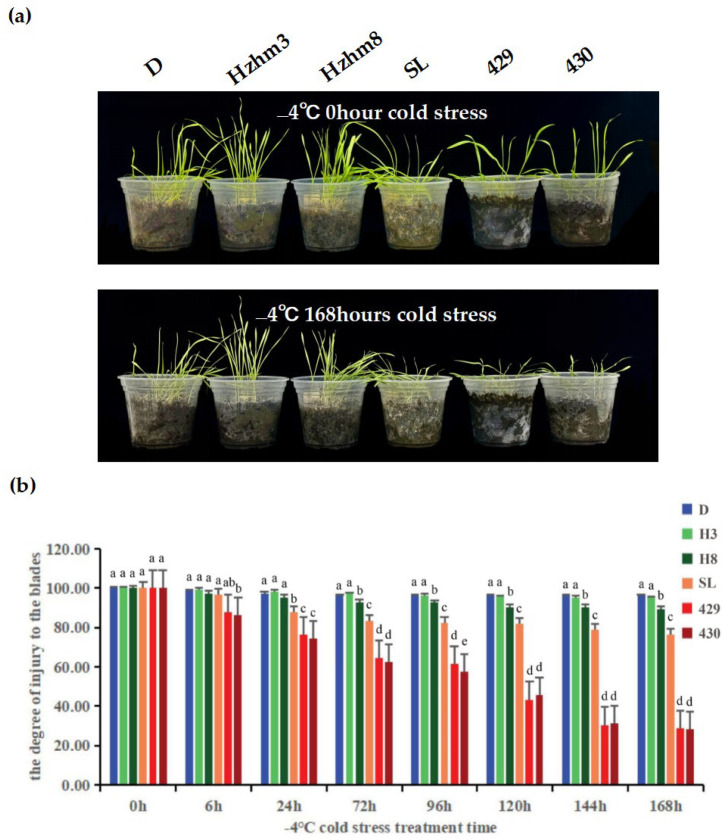
(**a**) is a phenotypic diagram showing the leaf changes of six rye varieties under different low-temperature stress treatments. The left and right sides of each sampling time image depict the leaf growth conditions of D, Hzhm3, Hzhm8, SL, 429, and 430, respectively; (**b**) is a bar chart showing the degree of leaf injury of the six ryes under different low-temperature stress treatments; the degree of leaf injury was measured by a leaf area meter, and statistical methods were used to analyze the significant differential changes. The treatment at 0 h served as the control group, while the treatments at other time points were the experimental groups. The letters such as a–c in the figure are the results marked based on statistical significance tests (such as multiple comparisons after analysis of variance). For different rye varieties under the same cold stress treatment time, if the letters marked above them are the same, it indicates that there is no significant difference in the degree of leaf injury among these varieties statistically; if the marked letters are different, it indicates that there are significant differences in the degree of leaf injury among these varieties statistically.

**Figure 7 cimb-47-00171-f007:**
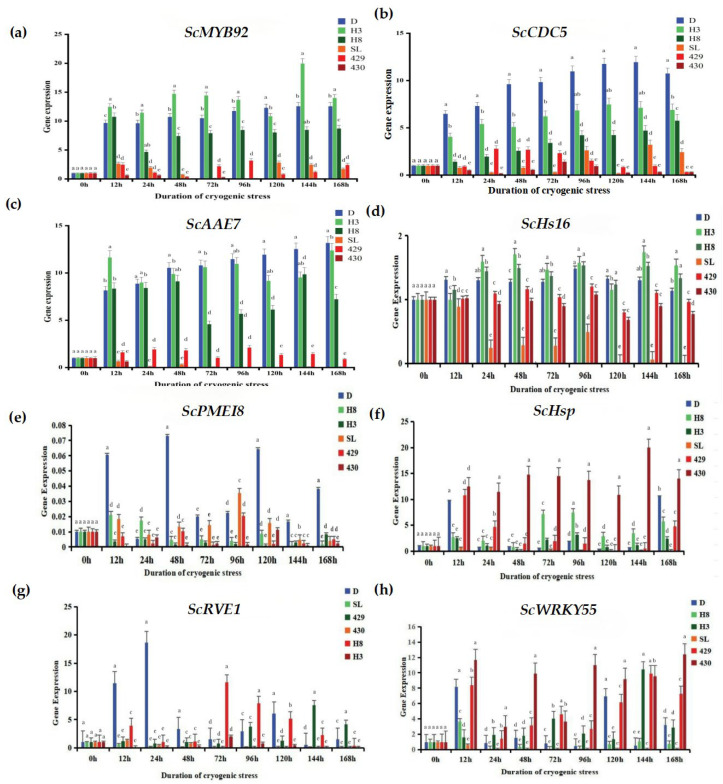
Changes in the expression levels of cold stress-related genes. (**a**) Changes in the expression level of the *ScMYB92* gene under different durations of cold stress treatment; (**b**–**h**) represent the changes in the expression levels of cold stress-related genes *ScMYB92*, *ScCDC5*, *ScAAE7*, *ScHs16*, *ScPMEI8*, *ScHsp*, *ScRVE1*, *ScWRKY55* determined by qRT-PCR. The treatment at 0 h served as the control group, while the treatments at other time points were the experimental groups. The letters such as a–c in the figure are the results marked based on statistical significance tests (such as multiple comparisons after analysis of variance). For different rye varieties under the same cold stress treatment time, if the letters marked above them are the same, it indicates that there is no significant difference in the degree of leaf injury among these varieties statistically; if the marked letters are different, it indicates that there are significant differences in the degree of leaf injury among these varieties statistically.

**Table 1 cimb-47-00171-t001:** Comparison groups of differentially expressed genes (DEGs) used in this study.

“Winter” Rye (D)	“Victory” Rye (S)	“Winter” Rye (D) and “Victory” Rye (S)
D-CK vs. D-06 h	S-CK vs. S-06 h	D-CK vs. S-CK
D-CK vs. D-24 h	S-CK vs. S-24 h	D-06 h vs. S-06 h
D-CK vs. D-24 h	S-CK vs. S-72 h	D-24 h vs. S-24 h
D-06 h vs. D-24 h	S-06 h vs. S-24 h	D-72 h vs. S-72 h
D-06 h vs. D-72 h	S-06 h vs. S-24 h	
D-24 h vs. D-72 h	S-24 h vs. S-72 h	

Note: In Table 1, “winter” rye is abbreviated as D, “victory” rye is abbreviated as S, CK represents the control group. Each entry represents a comparison between different time points of low-temperature stress treatments in rye samples, which were utilized for DEG tests to analyze gene expression changes.

**Table 2 cimb-47-00171-t002:** Changes in the quality of gene clusters of “winter” rye and “victory” rye under different cold stress durations.

Sample	Total Number	Total Length	Mean Length	N50	N70	N90	GC (%)
D_24 h_1	40,175	41,574,740	1034	1575	1057	467	50.44
D_24 h_2	34,376	34,736,024	1010	1550	1033	443	50.46
D_24 h_3	38,770	40,401,073	1042	1592	1073	470	50.33
D_06 h_1	39,250	43,674,044	1112	1686	1146	519	50.16
D_06 h_2	37,025	39,863,221	1076	1629	1108	496	50.76
D_06 h_3	39,291	43,907,388	1117	1698	1156	514	50.29
D_72 h_1	41,108	42,773,550	1040	1582	1063	468	49.92
D_72 h_2	44,285	47,994,481	1083	1664	1119	492	50.04
D_72 h_3	41,510	45,979,517	1107	1658	1137	518	50.23
D_CK_1	44,419	44,579,764	1003	1491	1024	463	49.93
D_CK_2	42,111	40,972,394	972	1486	991	426	50.11
D_CK_3	41,184	39,812,769	966	1391	971	454	49.94
S_24 h_1	32,014	30,522,460	953	1416	955	433	50.80
S_24 h_2	39,018	42,228,972	1082	1633	1117	497	50.30
S_24 h_3	42,691	44,002,527	1030	1512	1045	487	49.66
S_06 h_1	41,547	45,793,030	1102	1671	1138	503	49.78
S_06 h_2	42,705	46,312,310	1084	1655	1122	489	50.34
S_06 h_3	38,374	41,252,253	1075	1627	1100	487	50.26
S_72 h_1	37,107	37,889,005	1021	1524	1037	468	50.14
S_72 h_2	42,095	46,655,142	1108	1668	1139	516	50.00
S_72 h_3	49,982	55,017,680	1100	1662	1137	511	49.59
S_CK_1	49,382	51,749,070	1047	1518	1068	501	48.91
S_CK_2	42,719	40,688,665	952	1377	956	440	49.57
S_CK_3	47,671	47,862,601	1004	1434	1006	487	49.4
All-Unigenes	144,371	222,447,790	1540	2178	1570	850	49.38

Note: “D” stands for winter rye (“winter”) samples, and “S” for spring rye (“victory”) samples; “6 h”, “24 h”, “72 h” in sample names denote 4 °C low-temp stress treatment times (6, 24, 72 h). “CK” is the control group (no stress); “_1”, “_2”, “_3” are biological replicates. Total Number: total gene clusters in a sample. Total Length: gene cluster total length (bp). Mean Length: average gene cluster length. N50, N70, N90: gene assembly quality indicators (gene cluster lengths at 50%, 70%, 90% of total length). GC (%): GC content percentage in gene clusters.

**Table 3 cimb-47-00171-t003:** CDS sequences in candidate coding regions.

Total Number	Total Length	N50	N90	Max Length	Min Length	GC (%)
94,449	93,447,723	1263	477	14,682	297	53.76

**Table 4 cimb-47-00171-t004:** Summary table of notesE.

Values	Total	NR	NT	SwissProt	KEGG	KOG	Pfam	GO	Intersection	Overall
Number	144,371	114,619	122,993	81,513	86,479	81,249	80,546	80,822	49,045	129,186
Percentage	100%	79.39%	85.19%	56.46%	59.90%	56.28%	55.79%	55.98%	33.97%	89.48%

## Data Availability

The original contributions presented in this study are included in the article. Further inquiries can be directed to the corresponding author.

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
