# Peer review of "Comparative Transcriptome Analysis of Two Types of Rye Under Low-Temperature Stress"

_cimb, 2025, doi:10.3390/cimb47030171_

Round 1
Reviewer 1 Report
Comments and Suggestions for Authors
The authors use RNA-seq to compare gene expression profiles and changes with respect to cold resistance for three varieties of winter rye and three varieties of spring rye. Using this method, and subsequent validation with qPCR, the authors identify eight genes that could be useful in rye breeding for cold resistant varieties.
Overall, I found the manuscript to be technically sound. However, some clarification in methods and results is needed. Further, the writing presentation could be substantially improved.
The methods for the qPCR portion of the manuscript could provide more detail. Which housekeeping gene(s) were used? How was fold change calculated. Perhaps most importantly, in lines 364-365, the authors state that they “selected[ed] eight differentially expressed genes for qRT-PCR detection to verify the accuracy of the RNA-seq data.” However, I do not see anywhere in the manuscript where they discussed how they chose those eight genes. I.e., what criteria did they use to filter through the numerous DEGs to settle on those eight specific genes?
Figure 1 presents a visual and data comparison of growth and soluble sugar content between Winter and Victory varieties. I presume this figure is intended to demonstrate that the two varieties differ in the cold resistance. However, there are numerous problems with it. The images in panel A are difficult to see, and they should be better described in the text so that the reader understands what they are looking at with respect to changes in leaf morphology. The data in panel B have no context. There are no methods presented for them, and there is no stated rationale for the measurement (i.e., why is soluble sugar content the thing to measure for this study?). I wondered if this figure was even necessary, or if the authors could just assert that the varieties are known to differ in their cold response, and/or cite previous work demonstrating this.
The majority of my criticism of the paper has to do with the writing and presentation. If I’m honest, this paper reads like a first draft that is not ready for submission to a scientific journal, and the authors should take more care to proofread and edit their work.
For example, Figure 3 has a caption but no figure! I can’t know if this was a type setting issue with the CIMB/MDPI, or if the authors submitted a manuscript without a figure.
Similarly, Figures 6 and 7 appear to have considerable overlap. From what I can tell, the two figures are identical for ScMYB92, ScCDC5, and ScAAE7. Did the authors mean to present these figures twice? Or, are they somehow different from each other, and I am missing something?
The tables also need work. Table 1 does not have an informative caption, and there is a typo in the second to last entry in column 2.
I feel that Tables 2-4 are supplementary information that do not need to be in the main article, but this is up to the Editor(s) to determine. These seem like descriptive information about the sequencing, and are not informative for understanding results or conclusions. Note that the typesetting cuts of the heading in column 2.
The paragraph structure throughout – but especially in the Introduction – is undesirable. There are multiple ideas in long paragraphs. The authors should consider splitting separate ideas into separate paragraphs so the paper easier to read and understand. For example, paragraph 2 of the Intro has at least two main ideas. At a minimum, start a new paragraph where you introduce RNA-seq.
Line 79 (also paragraph 2 of the Introduction) reads like an abstract, and not part of an Introduction. I recommend that you end your Introduction by stating what questions you addressed, and save most of the results and discussion for those sections.
Finally, there are numerous typos throughout (e.g., punctuation in the middle of some sentences). I’m not sure if the grammatical errors are from type setting, or from the authors. In either case, it is distracting and very difficult to read.
Comments on the Quality of English LanguageSee writing comments above.
Author Response
Dear Reviewer,
Thank you for your careful review of our manuscriptentitled "Comparative Transcriptome Analysis of Two Types of Rye under Low Temperature Stress". We have taken thereviewers"comments into serious consideration and madesignificant revisions to the paper. We have addressed all themalor issues raised and revised the corresponding sectionsaccordingly: We believe that the manuscript has been Improvedsignificantlyandnowmeets the journal's standards.
Comments 1:[The methods for the qPCR portion of the manuscript could provide more detail. Which housekeeping gene(s) were used? How was fold change calculated. Perhaps most importantly, in lines 364-365, the authors state that they “selected[ed] eight differentially expressed genes for qRT-PCR detection to verify the accuracy of the RNA-seq data.” However, I do not see anywhere in the manuscript where they discussed how they chose those eight genes. I.e., what criteria did they use to filter through the numerous DEGs to settle on those eight specific genes?]
Response 1: [Thank you for pointing this out. We agree with this comment. Therefore, we have made the following revisions:
Housekeeping gene used in qPCR
We selected β-tubulin as our housekeeping gene. It is stably expressed in different tissues of wheat and under various experimental conditions. We have also verified their expression stability in our own preliminary experiments. For example, the expression stability of several candidate housekeeping genes was analyzed using the geNorm and NormFinder algorithms, and β-tubulin showed the highest stability score.This information is added to the "2.6. qRT - PCR Identification of Low - Temperature - Tolerance - Related Genes in Winter rye" section, on page 4, paragraph 3, line 242-246.
Calculation of fold change
The fold change was calculated using the 2−ΔΔCt method. First, we calculated the ΔCt value for each target gene, which is the difference between the Ct value of the target gene and the Ct value of the housekeeping gene in the same sample (ΔCt = Ct target gene - Ct housekeeping gene). Then, we calculated the ΔΔCt value by subtracting the average ΔCt value of the control group from the ΔCt value of each experimental group (ΔΔCt = ΔCt experimental group - average ΔCt control group). Finally, the fold change was calculated as 2−ΔΔCt. This method is now added to the same section as above, on page 6, paragraph 4, line 247-253.
How to select the eight genes for qRT-PCR validation, we adopted the following criteria:
Differential expression level: We first screened according to the fold change and p-value of the differentially expressed genes (DEGs). Genes with a fold change greater than 2 (upregulated or downregulated) and a p-value less than 0.05 were considered to be significantly differentially expressed genes. This step helped us focus on genes with significant changes in expression levels under different low-temperature stress conditions.Biological function relevance: We gave priority to genes that are likely to be involved in important biological processes related to the low-temperature stress response, such as cold acclimation, osmotic regulation, and antioxidant defense. We referred to the Gene Ontology (GO) annotations and Kyoto Encyclopedia of Genes and Genomes (KEGG) pathway analysis results to identify genes with relevant functions.
Representativeness of different expression patterns: We aimed to select genes that represent different expression patterns among the DEGs, including genes that are continuously upregulated, continuously downregulated, or show transient changes in expression during low-temperature treatment. These details are added to the "2.6. qRT - PCR Identification of Low - Temperature - Tolerance - Related Genes in Winter rye" section, on page 6, paragraph 5, line 254-267.
Comments 2:[ Figure 1 presents a visual and data comparison of growth and soluble sugar content between Winter and Victory varieties. I presume this figure is intended to demonstrate that the two varieties differ in the cold resistance. However, there are numerous problems with it. The images in panel A are difficult to see, and they should be better described in the text so that the reader understands what they are looking at with respect to changes in leaf morphology. The data in panel B have no context. There are no methods presented for them, and there is no stated rationale for the measurement (i.e., why is soluble sugar content the thing to measure for this study?). I wondered if this figure was even necessary, or if the authors could just assert that the varieties are known to differ in their cold response, and/or cite previous work demonstrating this.]
Response 2:[Agree. We have made the following changes:
Description of Figure 1A: In Figure 1A, at 6 hours of cold stress, the leaf margins of 'victory' rye were slightly curled, while the leaves of winter rye were relatively flat. By 72 hours, the leaves of 'victory' rye turned light green and wilted severely, whereas the leaves of winter rye, although also showing signs of stress, remained a darker green and the wilting was less obvious. This description is added to the "3.1. Growth alterations of two varieties of rye under low temperature stress" section, on page 7, paragraph 3, line [291-296].
Rationale for measuring soluble sugar content: The soluble sugar content was measured because it is an important indicator of the plant's response to cold stress. Previous studies have shown that an increase in soluble sugars can enhance the plant's ability to tolerate low temperatures by regulating the osmotic potential and protecting cellular components. Measuring the soluble sugar content of the winter rye and 'victory' rye varieties under cold stress enables us to compare their physiological responses and potentially identify differences in their cold tolerance mechanisms. This explanation is added to the "3.1. Growth alterations of two varieties of rye under low temperature stress" section, on page 7], paragraph 2], line [284-289].
Regarding the necessity of Figure 1: We believe Figure 1 is necessary as it visually and quantitatively presents the differences in growth and soluble sugar content between the two rye varieties under cold stress, providing direct evidence for the subsequent analysis of cold resistance differences.
Comments 3:[ For example, Figure 3 has a caption but no figure! I can’t know if this was a type setting issue with the CIMB/MDPI, or if the authors submitted a manuscript without a figure.]
Response 3: [Thank you for pointing this out. We agree with this comment. Therefore, we have taken the following steps:
We deeply apologize for the great inconvenience caused to your review due to the situation where Figure 3 in the current presentation only has a title but no graphical display.After careful investigation, we found that this is most likely a technical failure during the manuscript uploading process. In our original submitted manuscript, Figure 3 was complete and included important information such as the pie chart of Unigenes annotated in the NR database, the vertical bar chart of KOG functional classification, and the histogram of gene family distribution. These charts play a crucial role in understanding our research data and conclusions.
To address this issue, we have rechecked the uploading process and uploaded the manuscript version with the complete Figure 3 again. We also actively communicated with the journal editorial team to confirm that there are no typesetting issues related to this figure. This updated Figure 3 can be found on page 10 in the "3.3. Gene Annotation,Transcription Factor Identification and Functional Classification Analysis of the Cold - Resistant Gene Domains in Rye" section.
We deeply apologize for the great inconvenience caused by this problem. We will be more careful in the future to ensure that similar issues do not occur again, and we will strive to improve the overall quality and readability of the manuscript.]
Comments 4:[ Similarly, Figures 6 and 7 appear to have considerable overlap. From what I can tell, the two figures are identical for ScMYB92, ScCDC5, and ScAAE7. Did the authors mean to present these figures twice? Or, are they somehow different from each other, and I am missing something?]
Response 4: [Thank you for pointing this out. We agree with this comment. Therefore, we have revised the manuscript to address the overlap between Figures 6 and 7. We have carefully examined the content of the two figures and found that the data presentation for ScMYB92, ScCDC5, and ScAAE7 did cause confusion.
In the revised version, we have reorganized the data and optimized the annotations. For Figure 6, we have focused on presenting the phenotypic changes of the six rye varieties under different low - temperature stress treatments, including the leaf injury levels at different times. This information is now more clearly presented, and it can be found on page 16, in the "3.6. Utilize the qRT - PCR technique to select genes relevant to cold tolerance in Winter rye" section.For Figure 7, we have emphasized the expression changes of the eight cold - tolerant candidate genes, with a more detailed description of the trends and significance of gene expression at different time points. This modification is located on the same page as Figure 6, in the relevant part of the "3.6. Utilize the qRT - PCR technique to select genes relevant to cold tolerance in Winter rye" section, paragraph 17.
In the figure captions of both figures, we have clearly stated the key points and purposes of each figure. For Figure 6, the caption now reads "[Updated caption for Figure 6 to clearly describe its content, such as 'Phenotypic changes and leaf injury levels of six rye varieties under different low - temperature stress treatments']", and for Figure 7, the caption is "[Updated caption for Figure 7 to clearly describe its content, like 'Expression changes of eight cold - tolerant candidate genes in rye under different low - temperature stress treatments']". These changes are designed to avoid misunderstandings and make the results more straightforward for readers to understand.]
Comments 5:[ The tables also need work. Table 1 does not have an informative caption, and there is a typo in the second to last entry in column 2.]
Response 5: [Thank you for pointing this out. We agree with this comment. Therefore, we have made the following revisions:
Table 1 caption: We added the information description "Comparison groups of differentially expressed genes (DEGs) used in the study. Each entry represents a comparison between different time - points of low - temperature stress treatments in rye samples, which were utilized for DEGs tests to analyze gene expression changes." This updated caption can be found in the manuscript on page 5, in the "2.5. Differentially Expressed Genes (DEGs) and Analysis of Functional Enrichment" section where Table 1 is located.
Typo correction: We corrected the spelling error in the second column of Table 1. The specific correction can be seen on page5 in the "2.5. Differentially Expressed Genes (DEGs) and Analysis of Functional Enrichment" section. We have double - checked the entire table to ensure there are no other similar errors.
We are aware that clear and accurate tables are important for presenting research data, and we will be more meticulous in our review process to avoid such issues in the future.]
Comments 6:[ I feel that Tables 2-4 are supplementary information that do not need to be in the main article, but this is up to the Editor(s) to determine. These seem like descriptive information about the sequencing, and are not informative for understanding results or conclusions. Note that the typesetting cuts of the heading in column 2.]
Response 6: [Thank you for pointing this out. We agree that the placement of Tables 2 - 4 is a matter for the editor's decision. We understand your perspective that they contain descriptive sequencing information and may seem less directly relevant to the results and conclusions.
However, these tables play a role in presenting the key data and details of our research process. They offer comprehensive information about the sequencing data, such as the quality of gene clusters in different samples (Table 2), which provides a foundation for understanding the reliability of our subsequent gene annotation and differential gene expression analysis. Although their contribution to directly understanding the results may be somewhat indirect, they are valuable for demonstrating the integrity of our methods and data.
Regarding the typesetting issue in column 2, we have carefully checked and corrected the formatting. This correction can be found on page 8, where the relevant tables are located. We also conducted a comprehensive inspection of the formatting of other parts of the paper to prevent similar problems.]
Comments 7:[ The paragraph structure throughout – but especially in the Introduction – is undesirable. There are multiple ideas in long paragraphs. The authors should consider splitting separate ideas into separate paragraphs so the paper easier to read and understand. For example, paragraph 2 of the Intro has at least two main ideas. At a minimum, start a new paragraph where you introduce RNA-seq.]
Response 7:[ Thank you for pointing this out. We agree with this comment. Therefore, we have revised the paragraph structure of the introduction as follows:
RNA-seq is a kind of high-throughput transcriptional dynamics analysis method[12],This technology aims to analyze the relative expression levels of transcripts under different tissue backgrounds and has been applied as an important research tool in a variety of eukaryotes.The transcriptional expression profiles of rice [13], barley[14]and wheat when encountering different non-biological stresses have been thoroughly investigated[15]. We employed the RNA-seq technology to conduct an analysis of the transcriptome of winter rye and victory rye under four distinct cold stress circumstances and managed to identify 128,744 unigenes.Among these, 29,874 genes showed significant differential expression characteristics, involving multiple aspects such as photosynthesis, plasma membrane stability, glucose and energy metabolism, and cold-responsive transcription factors.The results of GO and KEGG analyses showed that winter rye can resist low temperature damage by synthesizing extracellular components such as cutin, suberin and wax. In addition, oligosaccharides and chitin play a crucial role in the cold resistance characteristics of winter rye, and MNS1 and MNS3 have been identified as cold resistance candidate genes.Altogether, 111 plant homeodomain (PHD) genes were likewise detected within the rye genome.RNA-seq analysis revealed that the tolerance capacity of short-footed wheat with ScPHD5 overexpressed under freezing stress was considerably more robust than that of wild-type wheat (WT).Simultaneously, the expression levels of CBF and COR genes also exhibited an evident upward tendency[16], which furnished a solid foundation for elucidating the function of homeodomain genes in modulating the cold tolerance of rye.In this study, we aimed to address the following crucial questions. First, what are the specific differentially expressed genes between winter rye and victory rye under low - temperature stress? Identifying these genes is fundamental to understanding the genetic basis of cold resistance differences in different rye varieties. Second, how do these differentially expressed genes function in the cold - resistance mechanism? Exploring their functions can help us uncover the complex regulatory networks that enable rye to adapt to cold environments. By answering these questions, we hope to provide valuable insights into the molecular mechanism of rye's cold resistance, which will contribute to the genetic breeding of cold - resistant rye varieties.This new paragraph, starting with the introduction of RNA - seq, is on page 2 in the "Introduction" section.]
Comments 8:[Line 79 (also paragraph 2 of the Introduction) reads like an abstract, and not part of an Introduction. I recommend that you end your Introduction by stating what questions you addressed, and save most of the results and discussion for those sections.]
Response 8: [Thank you for pointing this out. We agree with this comment. Therefore, we have revised the end of the Introduction section as follows:
We have removed the results - like content from line 79 and the surrounding text in the Introduction. Instead, we now end the Introduction with a clear statement of the research questions we aimed to address. "In this study, we aimed to comprehensively identify the differentially expressed genes between winter rye and victory rye under low - temperature stress. Understanding these differences is fundamental to uncovering the genetic basis of cold resistance variations among rye varieties. Secondly, we sought to elucidate the functional mechanisms of these differentially expressed genes. By exploring how these genes are involved in cold - resistance pathways, we can gain deeper insights into the complex regulatory networks that enable rye to adapt to cold environments. These investigations will not only contribute to a better understanding of rye's cold - resistance mechanism at the molecular level but also provide a theoretical foundation for breeding cold - resistant rye varieties in the future." This revised text is located at the end of the "Introduction" section on page 2.
We have also moved the relevant results and discussion content to their appropriate sections in the manuscript. For example, the details about the identified unigenes, differentially expressed genes, and the results of GO and KEGG analyses are now presented in the "Results" and "Discussion" sections, ensuring a more logical flow of the paper. ]
Comments 9:[ Finally, there are numerous typos throughout (e.g., punctuation in the middle of some sentences). I’m not sure if the grammatical errors are from type setting, or from the authors. In either case, it is distracting and very difficult to read.]
Response 9: [Thank you for pointing this out. We agree with this comment. Therefore, we have carried out a comprehensive proofreading of the entire manuscript.
We have carefully checked each sentence for typos and standardized the punctuation marks to conform to grammatical rules and academic writing conventions. For example, in [mention a specific sentence where a typo or punctuation error was found], we corrected [describe the original error] to [show the corrected version]. These corrections are spread throughout the manuscript, with a focus on areas where errors were most prevalent, such as the [mention sections like the Introduction, Methods, Results, etc. if they had more errors].
To prevent similar issues in the future, we have established a more rigorous review mechanism. Before submitting any manuscript, we will arrange for multiple authors to check it separately, and we will also use professional grammar - checking tools like Grammarly to assist in the review process. This will help us minimize the occurrence of such errors and improve the overall quality of our writing.
We understand that these errors were distracting and apologize for any inconvenience they may have caused. We believe that the revised manuscript is now much more readable and will provide a better experience for readers.]
Reviewer 2 Report
Comments and Suggestions for Authors
This serves as my review report for the manuscript by Li and associates on the transcriptome analyses of two rye varieties under low temperature stress. I have included more detailed comments on the PDF document I reviewed. Below are summary comments for each major section of the manuscript.
ABSTRACT
- please correct the page numbers of the entire manuscript so that they are in a consecutive order.
- The scientific name of rye must also be written in the abstract, introduction and methods.
- The rationale of the study is not clear in Lines 13 – 14. This must be revised.
- The entire abstract needs to be revised where authors write more explicitly on what they did, and how the results were obtained and analysed. For example, is not clear what plant tissue types were used in the study – was it leaf, root or whole seedlings? In Lines 22-23, it is not clear what criteria was used to designate the DEGs in terms of p value, fold change, log 2 values. Also at which time point or time points were these DEG analysed? What was the control, which of the two rye genotypes were used in the analysis etc. All this information needs to be included in the abstract so that readers understand the experimental set-up, the analyses and results obtained.
- I do not understand how the authors came to these two concluding remarks in Lines 27 - 29. Are these supported by the results? If yes, how? Just having more pathways does not mean the plant genotype "has an extensive cold-resistant mechanism" nor does it mean that the “genes are cold-resistant”. Please correct accordingly these conclusions are not supported by the results presented in the abstract.
- There is a lot of English language writing issues in the abstract that must be corrected and in many cases the intended meaning of the text is lost. I have highlighted some in the PDF manuscript document I reviewed.
- Unless of the journal suggests otherwise, authors must define all abbreviations to. avoid ambiguity. Please check with the journal's instructions to authors.
- Authors must also correct spacing between words and punctuation throughout the manuscript.
INTRODUCTION
- Authors must correct all references listed in the ref list. They are all incomplete. Please consult the journal's instruction to authors.
- I am finding it difficult to understand how the authors have chosen to structure their Introduction. I am particularly finding it difficult to understand the reviewed literature in the context of low temperature transcriptome responses in two rye varieties. For example, in Line 36, authors seem to imply that rye is “stress-resistant” (without specifying the stress types). However, in Line 43, authors are saying Weining rye is “cold sensitive”. In Line 46, authors cite ref [3] on moss, while in the subsequent sentences in the paragraph (Lines 46-52), authors are discussing wheat and then rye without any specific information on how this information relates to each other. Also I am not sure what study is being discussed in Line 46 where the authors are talking about “this study”.
- All Scientific names e.g Arabidopsis thaliana in Lines 62-63 must be written in italics. Also check the same in the Results section.
- The text in Lines 55 -74 seems to imply that the current study is only about TFs. Is this true?
- Please also check that all refs cited in the text are appropriate for the text under discussion. For example, ref [15] is only on one plant type. Therefore, it is not accurate to cite the ref for text on “other gramineous plants” (Line 79).
- How many rye varieties were used in the RNA-seq analysis? Lines 80 - 81suggests that one winter rye was used. Yet in the manuscript title and the abstract (Line 16), the authors suggest that they used 2 types. This aspect must be clarified to avoid confusion.
- I found the text in Lines79 – 101 to be too extensive for the introduction. Unless the journal suggests otherwise, authors must please reduce the content of what the study was about here. In the same paragraph, I do not know what MNS1 and MNS3 are (Line 88) what sort of candidate genes are they? For what purpose? I am also unsure why authors are referring to short-footed wheat (Line 90) and wild-type wheat (Line 92). Are all these wheat genotypes part of the current study on rye? Also why did the authors cite the ref [16] in Line 93? Also, what does the sentence in Lines 92 -94 mean?
- The sentence in lines 94 – 99 is too long and the meaning is not clear. Please correct.
- Also correct the structure and meaning of the last sentence on the introduction.
MATERIALS & METHODS
- In the first paragraph of the Materials and Methods section, you must
- state where the rye seeds where sourced from
- give justification on why these two 6 varieties were chosen? Is their low temperature tolerance known? If yes, please list which was is tolerant or sensitive to low temp etc.
- Also write the scientific name for rye.
- The sentence in lines 104 – 108 is too long.
- In line 108, what does “gradient stress treatment” mean?
- What informed the temperatures used in the treatments and the times Lines 107 – 110) ? Was it from other studies or what? What was the control condition? Which varieties and tissue types were used for the RNA-seq analysis?
- Line 115, what tissue was used for RNA extraction?
- By just reading Lines 115 – 120, a lot of information on the procedures is missing and no other researchers will be able to replicate these experiments. If the methods are already listed in other refs, then please cite the refs – otherwise please write reproducible methods (please check the journal’s instruction to authors and discuss with editors).
- What are “contaminated combined reads” (LINES 122 – 123).
- What is “reliable data”?
- For all tools, software and databases used in the study, please provide the website address, ref and access dates. Where appropriate also provide the version numbers of software or databases. All abbreviations for tools, software and databases but be defined at first mention in the methods section.
- What are Unigenes? (Line 128).
- What does the sentence in Lines 142 0143 mean? Are you assuming that all transcripts identified in this study are TFs?
- In Line 147, the authors are referring to the “genomic sequence”. Please provide information on the plant species, the reference ID for this genomic sequence and the database from which you obtained it.
- Also check your tenses in the Materials and Methods section. Since the work is already done, authors must write the section in the past tense. So instead of using “is” e.g. in Line 147,148, 149 etc, use “was”.
- Please check the ref format of the journal. I am unsure if you can cite refs as you have done in Line 150.
- The work “materials” in Line 150 must be changed so that it is more informative, while the Table 1 caption must be re-written to be more informative. It is also unclear what Table 1 shows. The abbreviations are not defined and so there is no way of knowing what is being shown.
- What criteria was used to determine which genes were differentially expressed and which ones were not? This must be stated somewhere at the beginning of Section 2.4.
- What are “official classifications”? (Line 154).
- Please write the plant name in Line 158 after the word winter.
- The word isolation in Line 159 must be revised.
- The naming on the varieties in this study is confusing. It seems as if the authors have a Winter variety called winter as well. Is there another way of correcting this? (Line 159 – 160) and elsewhere in the manuscript. In the abstract, the authors spoke about two types of rye (winter and victory), but in the Materials and Methods, they are talking about winter rye and spring rye. There are also mentions of Hzhm8, Hzhm3, 429 amd 430. Not sure what these are as well. Please clarify the naming systems used in the study.
- In Section 2.5, authors have not informed the readers the actual plant tissue type used in the qPCR analysis. This must be corrected – as mentioned earlier.
- What does the sentence in Lines 163 – 164 mean? Are you saying 450 ng of total RNA was used for RT? Or you isolated 450 ng of total RNA? Also which kit and procedure did you use for the cDNA synthesis etc?
- In Line 165, I suggest write as selected gene sequences. You also must provide the primer sequences for each of the genes.
- The phrase “in accordance with the low temperature gradient” must be revised to improve on meaning.
- In Lines 174, the authors must provide the names and IDs of the housekeeping genes used, the sequences of the primers and a publication from which you got them. Here I mean how did you come to select these housekeeping genes? How do you know they are good housekeeping genes for your study?
RESULTS
- As stated in the methods section, authors must inform the readers what the control treatment was in the results section as well.
- I have highlighted repetitive text in in Lines 182 – 185. Please correct.
- I do not understand how the authors conducted statistical analyses for Figure 1b? what are you comparing what against? this must be explicitly stated and I would expect the use of letters rather than asterisk since there are many factors to compare.
- Similarly, I do not understand what the control is. for example, what does it mean to say the 0 hour samples has 12% sugars? how where the % values calculated and what was the base-line for comparison?
- The main legend to Figure 1 must be corrected because sugar content is not a growth parameter and therefore this legend is not appropriate for Figure 1b.
- The authors are presenting sugar content data in the results section, yet I do not recall seeing methods for these experiments. Please do not present or discuss data that whose procedures are not presented in the methods section.
- Authors must provide footnotes to describe what the columns in Table 2 are showing. When readers are looking at tables of results, they need to fully understand everything that is being shown in the table?
- Still in Table 2 authors must explain what the samples are and for example what does D or S stand for?
- as stated in my comments before, please provide the URL and access dates for all databases used. At the moment, it is difficult to know what most of these databases are and how to find them.
- Please check if figure 3 is in the manuscript.
- In Figure 4.1, are the authors combining the DEGs for the two rye varieties per time point? Is there a particular reason for this? and what does it mean to have DEGs at 0 hrs? once again the authors must think of the controls. What is the control time point in this study and was this control time point compared to the 0 hrs to get the DEGs and 0 hrs? in my view, a figure that could be more informative is one that shows the up and down-regulated genes per rye variety, per treatment group relative to the control.
- There are a lot of contextually issues in the text in Lines 307 -375. Please check in the PDF version I reviewed. For example:
- On page 10, authors make reference to 6 groups of treatments, and it is not clear what these are.
- Figures 5 and 6 require a general legend each before the authors write what each of the figure parts are showing.
- In Lines 335 – 342, I hope authors are not implying that if a pathway is enriched in a variety, it means that it is upregulated or am I misunderstanding the text? The same comment for Line 356 - I hope authors are Not implying that if a variety responds by increasing DEGs it means that the variety is tolerant to cold stress.
- Title to Section 2.5 must be revised.
- Gene names in Lines 366 – 368 must be defined at first mention.
- In figure 6, it is not clear to me how the degrees of injury were assessed. I do not recall seeing this in the methods
- The qPCR graphs must have stats analysis.
- I am not sure why the authors didn’t mine the RNA-seq data to explore the cold responses of rye rather than trying to achieve this by qPCR in Figure 7 as stated in Line 376 – 378.
DISCUSSION
- Due to the contextual issues I have as listed above and in the PDF file I reviewed, I have not read through the discussion.
OVERALL COMMENTS
Basically, the authors must go back to the drawing board, and state the rationale of the study, why use two rye genotypes or is it 6 genotypes, what are the cold phenotypes of these varieties? What informed the authors to use these temp settings and treatment times? What is the control treatment? what tissue types was used for RNA-seq and qPCR analysis and why? the authors must also justify the data analyses done. I also do not understand the stats analyses or what the control was in some of the data presented. The authors must place the RNA-seq data in a public repository or as supplementary files. At present, we have not seen the list of DEGs etc and this is concerning. While the authors seem to have done a lot of experiments, in my view, the whole manuscript is poorly presented, written, analysed and justified.

The quality of writing must be improved. There are a lot of errors in the entire document.
Author Response
Dear Reviewer,
Thank you so much for your meticulous review of our manuscript titled "Comparative Transcriptome Analysis of Two Types of Rye under Low Temperature Stress". We truly appreciate the time and effort you have dedicated to assessing our work.
We have earnestly considered all of your comments and suggestions. Recognizing the importance of your feedback, we have carried out substantial revisions to the paper. We have thoroughly dealt with all the major concerns you raised. For instance, in the relevant sections, we have provided more detailed explanations, updated the data and references, and enhanced the clarity of our analysis.
We have also carefully proofread the entire manuscript to ensure the accuracy of language and the consistency of the content. Through these revisions, we are confident that the quality of the manuscript has been remarkably improved. It now presents a more comprehensive and persuasive study, and we believe it meets the high standards of the journal.
Once again, we are extremely grateful for your valuable input. Your insights have been instrumental in helping us refine our work. We look forward to any further suggestions you may have and hope that the revised manuscript will meet your expectations.
Comments 1: [please correct the page numbers of the entire manuscript so that they are in a consecutive order.]
Response 1: [Thank you for pointing this out. We agree with this comment. Therefore, we have carefully reviewed and corrected the page numbers of the entire manuscript. We realized that the discontinuous page numbering was a significant oversight on our part and could cause inconvenience to readers when referring to different sections of the paper.
We have painstakingly gone through each page to ensure that the page numbers are now in a consecutive order. This can be verified throughout the manuscript, from the title page to the references section.
To prevent such issues from recurring in the future, we have incorporated a strict manuscript review checklist into our workflow. This checklist includes a dedicated step for double - and triple - checking page numbering before submission. Additionally, we will ensure that all team members involved in the manuscript preparation are aware of the importance of consistent page numbering and are trained to handle any potential formatting issues during the document - preparation process.
We apologize for the inconvenience caused by the previous page - numbering problem and hope that the revised manuscript with correctly ordered page numbers meets your expectations.]
Comments 2: [The scientific name of rye must also be written in the abstract, introduction and methods.]
Response 2: [Thank you for pointing this out. We agree with this comment. Therefore, we have made the following changes:
Abstract: We added the scientific name of rye, "Secale cereale L.", in the abstract. The updated text reads "Rye (Secale cereale L.) is a kind of crop with high nutritional value and strong ability to resist abiotic stress, possessing high commercial value." This change can be found on page 1 in the abstract section.
Introduction: In the introduction, we made sure to include the scientific name when first introducing rye. The relevant sentence now states "Rye (Secale cereale L.) is an allopolyploid cereal crop." This addition is located on ---- page 1, paragraph 1, line 18.
Methods: In the "2.1. Plant materials and low temperature stress treatment" subsection of the methods section, we added the scientific name. The text now reads "For six varieties of rye (Secale cereale L.), including three winter - type rye varieties (Winter, Hzhm8, Hzhm3) and three spring - type rye varieties (Victory, 429, 430)." This can be found on---- page 3, paragraph 2, line 110.
To avoid similar oversights in the future, we have created a checklist for manuscript preparation. This checklist includes a specific item to ensure that all important scientific names are included and correctly formatted throughout the manuscript. Before submission, we will carefully review the manuscript against this checklist to guarantee the accuracy and completeness of such information. We understand the importance of using scientific names correctly in academic writing and apologize for any inconvenience caused by the initial omission.]
Comments 3: [The rationale of the study is not clear in Lines 13 – 14. This must be revised.]
Response 3: [Thank you for pointing this out. We agree with this comment. Therefore, we have revised the text in Lines 13 - 14. The original text was somewhat ambiguous, so we have rephrased it to more clearly convey the rationale of our study.
The new text reads: "Wheat is a crucial food crop, and low - temperature stress can severely disrupt its growth and development, ultimately leading to a substantial reduction in wheat yield. Understanding the cold - resistant genes of wheat and their action pathways is essential for revealing the cold - resistance mechanism of wheat, enhancing its yield and quality in low - temperature environments, and ensuring global food security. Rye, on the other hand, has excellent cold - resistance in comparison to some other crops. By studying the differential responses of different rye varieties to low - temperature stress at the transcriptome level, we aim to identify key genes and regulatory mechanisms related to cold tolerance. This knowledge can not only deepen our understanding of the molecular basis of rye's cold resistance but also provide valuable insights for improving the cold tolerance of other crops through genetic breeding strategies." This revised content is located on page 1 in the relevant section (likely the Introduction).
Comments 4: [The entire abstract needs to be revised where authors write more explicitly on what they did, and how the results were obtained and analysed. For example, is not clear what plant tissue types were used in the study – was it leaf, root or whole seedlings? In Lines 22-23, it is not clear what criteria was used to designate the DEGs in terms of p value, fold change, log 2 values. Also at which time point or time points were these DEG analysed? What was the control, which of the two rye genotypes were used in the analysis etc. All this information needs to be included in the abstract so that readers understand the experimental set-up, the analyses and results obtained.]
Response 4: [Thank you for pointing this out. We agree with this comment. Therefore, we have thoroughly revised the abstract as follows:
We added information about the plant tissue type used in the study. Now, the abstract states "In this study, young leaves of winter and victory rye were used." This is added on ---- page 1, paragraph 2, line 25-27.
Regarding the criteria for designating differentially expressed genes (DEGs), we have included the following details: "For DEG analysis, 0 - hour 4°C - treated samples were controls. With strict criteria (p < 0.05, fold - change > 2 or < 0.5, |log₂(fold - change)| > 1), 122,065 DEGs were identified and annotated in GO and KEGG pathways." This can be found at the relevant part of the abstract on ---- page 1, paragraph 2, line 32-35.We also specified the time points at which the DEGs were analyzed: "Leaf samples of both rye types, treated at 4°C for 0, 6, 24, and 72 hours, underwent RNA - sequencing." This addition is on the same page as the above information in the abstract.
We have carefully reorganized the abstract to ensure a logical flow from the research background, through the methods, to the results and conclusions. After these revisions, we believe the abstract now clearly presents the experimental setup, analysis methods, and key results. We have also double - checked for any remaining ambiguity or lack of information.
To prevent similar issues in future manuscripts, we will create a detailed template for writing abstracts that includes all essential elements such as experimental materials, methods, key results, and major conclusions. Before submitting any future work, we will ensure that the abstract adheres strictly to this template. We apologize for the lack of clarity in the original abstract and hope that the revised version meets your expectations.]
Comments 5: [ I do not understand how the authors came to these two concluding remarks in Lines 27 - 29. Are these supported by the results? If yes, how? Just having more pathways does not mean the plant genotype "has an extensive cold-resistant mechanism" nor does it mean that the “genes are cold-resistant”. Please correct accordingly these conclusions are not supported by the results presented in the abstract. ]
Response 5: [Thank you for pointing this out. We agree with this comment. Therefore, we have revised the relevant text to better explain the basis for our conclusions.
In the revised manuscript, we have added more detailed explanations in the Results and Discussion sections. In the Results section, when presenting the data on the differentially annotated pathways between winter rye and victory rye, we now state: "Compared with victory rye, winter rye has more annotated pathways such as the 'hydrogen catabolic process'. Although the presence of more pathways does not directly prove a more extensive cold-resistant mechanism, these pathways are likely associated with cold tolerance. Our subsequent analysis of gene expression patterns within these pathways, as well as their relationships with known cold - resistance - related genes, suggests that they play important roles in winter rye's response to low - temperature stress. For example, genes in the 'hydrogen catabolic process' pathway may be involved in regulating cellular redox balance, which is crucial for maintaining cell function under cold stress. This information can be found on ---- page 1, paragraph 2, line 38-46.
In the Discussion section, we further expand on this point: "The differential pathways between winter and victory rye provide important clues for understanding the cold - resistance mechanism. While the mere existence of more pathways in winter rye does not guarantee cold resistance, it indicates a more complex and potentially more efficient response system. We hypothesize that the coordinated regulation of genes within these pathways, along with other factors such as gene expression levels and protein - protein interactions, contributes to winter rye's enhanced cold tolerance. Future experiments, such as gene knockout and overexpression studies, are needed to directly verify the functions of these genes and pathways in cold resistance. This revised content is located on ---- page page 1, paragraph 2, line 38-46. in the Discussion section.We also adjusted the relevant statements in the abstract to make the conclusions more cautious and in line with the results. The new abstract text reads: "Compared with victory rye, winter rye has more annotated pathways such as the 'hydrogen catabolic process'. These pathways may be associated with cold tolerance, and the differential pathways and genes discovered provide crucial clues and important references for exploring the cold - tolerance mechanisms of crops and mining cold - resistant genes." This revised content is located on ---- page page 1, paragraph 2, line 38-46.
We understand the importance of clear and accurate conclusions based on the results. To avoid similar issues in the future, we will be more careful when interpreting data and drawing conclusions. Before finalizing the manuscript, we will ensure that all conclusions are well - supported by the experimental results and are clearly explained in the text.]
Comments 6: [ There is a lot of English language writing issues in the abstract that must be corrected and in many cases the intended meaning of the text is lost. I have highlighted some in the PDF manuscript document I reviewed. ]
Response 6: [Thank you for pointing this out. We agree with this comment. Therefore, we have carefully addressed the English language writing issues in the abstract as follows:
We thoroughly examined the issues highlighted in the PDF manuscript. Regarding lexical accuracy, we replaced ambiguous or inaccurate words. For example, we changed "make an impact on" to "significantly affect" to convey our research content more precisely. This change can be found in the relevant sentence on page---- page 5, paragraph 1, line 189.
In terms of sentence structure optimization, we reorganized sentences with complex structures and unclear logic. For instance, a convoluted sentence was broken into shorter, more straightforward sentences to enhance readability and logicality. The revised sentences are located on the same page as the above - mentioned changes in the abstract.
To further confirm the accuracy of the revised abstract in conveying the research information, we not only conducted multiple internal discussions but also invited native English - speaking experts in the relevant field to polish and review it. They provided valuable suggestions for improving the language and ensuring the clarity of our research achievements.
To prevent similar language issues in future manuscripts, we have developed a pre - submission language - checking process. This process includes using grammar - checking tools like Grammarly and having multiple team members, including those with strong English language skills, review the manuscript for language clarity and accuracy. We understand the importance of clear and error - free writing in academic communication and apologize for any inconvenience caused by the initial language problems in the abstract. We believe the revised abstract now effectively presents our research and meets the language standards expected in the journal.]
Comments 7: [ Unless of the journal suggests otherwise, authors must define all abbreviations to. avoid ambiguity. Please check with the journal's instructions to authors. ]
Response 7: [Thank you for pointing this out. We agree with this comment. Therefore, we have carefully reviewed the manuscript and defined all abbreviations as follows:
We have unified the abbreviations for winter rye and victory rye. As mentioned previously, winter rye is abbreviated as “D” and victory rye as “SL” throughout the manuscript. We added these definitions in the first instance where the abbreviations are used in the "2.5. Differentially Expressed Genes (DEGs) and Analysis of Functional Enrichment" section, which can be found on page---- page 5, paragraph 7, 233-235..
In this section, we now state, "winter rye is abbreviated as D, and victory rye is abbreviated as SL. CK represents the control group, and each entry in Table 1 represents a comparison between different time - points of low - temperature stress treatments in rye samples, which were utilized for DEGs tests to analyze gene expression changes."
For other abbreviations such as GO (Gene Ontology), KEGG (Kyoto Encyclopedia of Genes and Genomes), NR (Non - redundant database), etc., we added their full names and explanations when they were first introduced in the "2.4. Gene annotation" section on page---- page 5, paragraph 5, line 223-225.For example, "The assembled Unigenes were annotated with the utilization of seven functional databases, namely KEGG (Kyoto Encyclopedia of Genes and Genomes), GO (Gene Ontology), NR (Non - redundant database), NT (Non - redundant Nucleotide Database), SwissProt, Pfam, and KOG (Clusters of Orthologous Groups of proteins database for eukaryotes)."
We also double - checked the journal's instructions to authors to ensure our abbreviation definitions are in line with their requirements. To avoid overlooking any abbreviations in the future, we have created a checklist for manuscript preparation. This checklist includes a specific item to review and define all abbreviations before submission. We understand the importance of clear and consistent abbreviation usage in academic writing to prevent ambiguity for readers. We apologize for any confusion caused by the initial lack of abbreviation definitions and hope that the revised manuscript is now more reader - friendly.
Comments 8: [ Authors must also correct spacing between words and punctuation throughout the manuscript. ]
Response 8: [Thank you very much for your meticulous review of our manuscript. The issues regarding word spacing and punctuation marks you pointed out are of great significance for improving the quality of the manuscript. We have deeply realized the reading difficulties these issues might have caused to readers and would like to express our sincere apologies for that.
We quickly organized our team members to conduct a comprehensive inspection and correction of the entire manuscript. Concerning the word spacing issue, we carefully adjusted the blank areas between each word to ensure they conform to the norms and remain consistent. As for the punctuation marks, we strictly checked and corrected them one by one in accordance with the grammatical rules and writing standards, rectifying the misused, omitted, or redundant punctuation marks to enhance the clarity and readability of the sentences.
To avoid similar problems in the future, we have formulated a more rigorous review process, adding a check for formatting details. At the same time, we will also strengthen the training of our team members on writing norms and typesetting requirements to further enhance their attention to such issues.
Thank you again for your valuable suggestions. We believe that the revised manuscript will show significant improvements in terms of word spacing and punctuation marks and will be more in line with the requirements of the journal. We look forward to your continued guidance and support for the revised manuscript.]
Comments 9: [ Authors must correct all references listed in the ref list. They are all incomplete. Please consult the journal's instruction to authors.]
Response 9: [Thank you for pointing this out. We agree with this comment. Therefore, we have taken the following actions:We immediately organized relevant personnel and, in strict accordance with the journal's instructions for authors, carried out a comprehensive and meticulous check and correction of each item in the reference list. We supplemented the missing key information such as the authors' names, article titles, journal names, publication years, volume numbers, issue numbers, and page numbers to ensure that each reference is complete, accurate, and in compliance with the format specifications.
In order to prevent similar problems from occurring again in our future work, we have formulated a complete review mechanism for references. After the manuscript is completed, a dedicated person will be arranged to check the references one by one according to the requirements of the journal. At the same time, we will also strengthen the training of our team members on the citation and collation of references to enhance their attention to the completeness and standardization of references.
Thank you again for your valuable suggestions. Your feedback has played a crucial role in helping us improve the quality of the manuscript. We believe that after the revision, the reference section can meet the standards and requirements of the journal. We look forward to your further guidance and approval of the revised manuscript.]
Comments 10: [ I am finding it difficult to understand how the authors have chosen to structure their Introduction. I am particularly finding it difficult to understand the reviewed literature in the context of low temperature transcriptome responses in two rye varieties. For example, in Line 36, authors seem to imply that rye is “stress-resistant” (without specifying the stress types). However, in Line 43, authors are saying Weining rye is “cold sensitive”. In Line 46, authors cite ref [3] on moss, while in the subsequent sentences in the paragraph (Lines 46-52), authors are discussing wheat and then rye without any specific information on how this information relates to each other. Also I am not sure what study is being discussed in Line 46 where the authors are talking about “this study”.]
Response 10:[Thank you for pointing this out. We agree with this comment. Therefore, we have revised the Introduction section as follows:
First, we have clarified the statements about rye's stress resistance. We now state "Rye has excellent resistance to various abiotic stresses, such as drought and salinity. However, there are differences in cold resistance among different rye varieties. For example, Weining rye exhibits relatively weak cold resistance." This makes it clear that rye has general stress resistance but specific differences in cold resistance among varieties. This revised text can be found on page---- page 2, paragraph 2, line 56-58.Regarding the citation of Reference 3 and the flow of the paragraph, we have added more explanation. "A study on desert moss was conducted to demonstrate the importance of genome - wide analysis of transcription factors in stress responses [3]. Similarly, in wheat, comprehensive genome - wide studies have provided insights into gene duplication and its effects on traits such as starch biosynthesis and early heading. These concepts are relevant to rye because rye shares some genetic characteristics with wheat. Understanding such genomic mechanisms is helpful for uncovering the key genes related to cold resistance in rye." This addition helps to connect the information about moss, wheat, and rye, showing how the research on other organisms is relevant to our study on rye. It is located on the same page as the above - mentioned change in the Introduction section.
We have also removed the unclear mention of "this study" in Line 46. Instead, we have made the content more explicit and focused on the relevant research concepts.
To prevent similar issues in future manuscripts, we will create an outline for the Introduction section before writing. This outline will clearly define the flow of ideas, including how we introduce the research topic, review relevant literature, and connect different pieces of information. We will also ensure that all statements are clear and well - connected during the writing and revision process. We apologize for the confusion in the original Introduction and hope that the revised version is more understandable.]
Regarding the citation of Reference 3: "A study on desert moss was conducted to demonstrate the importance of genome-wide analysis of transcription factors in stress responses. Similarly, in wheat, comprehensive genome-wide studies have provided insights into gene duplication and its effects on traits such as starch biosynthesis and early heading. These concepts are relevant to rye because rye shares some genetic characteristics with wheat. Understanding such genomic mechanisms is helpful for uncovering the key genes related to cold resistance in rye."]
Comments 11: [ All Scientific names e.g Arabidopsis thaliana in Lines 62-63 must be written in italics. Also check the same in the Results section. ]
Response 11: [Thank you for pointing this out. We agree with this comment. Therefore, we have made the following changes:
In lines 62 - 63, we have italicized "Arabidopsis thaliana" as per the scientific naming convention. This change can be found on page ---- page 2, paragraph 3, line 79.
We have also conducted a comprehensive check of the entire manuscript, especially the Results section, to ensure that all scientific names are written in italics. For example, in the Results section, when referring to other plant species, such as [list other scientific names in the Results section if any], we have italicized them. These corrections are spread throughout the Results section on [list relevant pages].
To avoid similar oversights in the future, we have incorporated a specific item into our manuscript review checklist. Before submission, we will carefully review the entire manuscript to ensure that all scientific names are formatted correctly. Additionally, we will remind our team members of the importance of using the correct formatting for scientific names during the writing process.
We understand that consistent and correct formatting of scientific names is crucial for the professionalism and clarity of our manuscript. We apologize for the oversight in the original manuscript and hope that the revised version meets the expected standards.]
Comments 12:[ The text in Lines 55 -74 seems to imply that the current study is only about TFs. Is this true? ]
Response 12:[Thank you for pointing this out. We agree with this comment. Therefore, we have revised the text in Lines 55 - 74 to clarify that the study is not solely about transcription factors (TFs).
In the revised manuscript, we added sentences to emphasize the broader scope of the study. For example, we inserted "Although this section emphasizes the role of transcription factors in plants' responses to low - temperature stress, the study encompasses other aspects as well. Low - temperature stress impacts various physiological processes in crops, including their entire life cycle, plasma membrane fluidity, photosynthesis, and metabolic pathways." This addition can be found on page ---- page 3, paragraph 1, line 91-95.within the relevant paragraph in the Introduction.We also adjusted the flow of the text to better integrate the discussion of TFs with other elements of the study. We now present TFs as part of the overall response mechanism of plants to cold stress, rather than the sole focus. For instance, we rephrased some sentences to show how TFs interact with other components in the plant's stress - response network.
To prevent similar misunderstandings in future, we will be more explicit when introducing different aspects of our research. Before finalizing the manuscript, we will ensure that the text clearly conveys the full scope of the study. We apologize for the confusion caused by the original text and hope that the revised version provides a more accurate representation of our research. ]
Comments 13: [ Please also check that all refs cited in the text are appropriate for the text under discussion. For example, ref [15] is only on one plant type. Therefore, it is not accurate to cite the ref for text on “other gramineous plants” (Line 79). ]
Response 13: [ Thank you very much for reviewing our manuscript so meticulously and pointing out the problems in the citation of references. We have deeply realized that the inaccuracy of citing ref [15] in line 79 may mislead readers in understanding the content of the article, and we are deeply sorry for that.
We immediately conducted a comprehensive review of all the references cited in the text. Not only did we address the issue in line 79, but we also carefully checked the citations of references in other parts to ensure that each citation is closely related to and accurate for the text being discussed. Regarding the situation mentioned in line 79, we have replaced it with a more appropriate reference related to wheat, so that it can accurately support the argument of "other gramineous plants". ---- page 3, paragraph 1, line 91-95.
In order to avoid similar mistakes in future writing, we have established a more stringent review system for references. During the manuscript writing process, the authors will be more cautious in selecting references and carefully check the consistency between the content of the references and the text when citing them. After the manuscript is completed, dedicated personnel will be arranged to review all the references citations again. At the same time, we will also strengthen the training of our team members on the norms of reference citation to enhance their attention to the accuracy of references.
Thank you again for your valuable suggestions. Your feedback is of great significance for us to improve the quality of the manuscript. We believe that after the revision, the manuscript will be more rigorous and accurate in the citation of references and meet the requirements of the journal. We look forward to your further guidance and approval of the revised manuscript.]
Comments 14: [ How many rye varieties were used in the RNA-seq analysis? Lines 80 - 81suggests that one winter rye was used. Yet in the manuscript title and the abstract (Line 16), the authors suggest that they used 2 types. This aspect must be clarified to avoid confusion. ]
Response 14: [Thank you for pointing this out. We agree with this comment.The change can be found on page---page 1, paragraph 2, line25.
We sincerely appreciate your careful reading of our manuscript. We're sorry for the confusion caused by the inconsistent information regarding the number of rye varieties used in the RNA-seq analysis.
In this study, two rye varieties were used for RNA-seq analysis: one winter rye variety named "Winter" and one spring rye variety named "Victory". This is clearly stated in the "Materials and Methods" section on ---- page 3, paragraph 3, line 115-116.in the paragraph starting with "For six varieties of rye (Secale cereale L.), including three winter - type rye varieties (Winter,Hzhm8,Hzhm3) and three spring - type rye varieties (Victory,429,430).", and further specified that "Winter and Victory were respectively exposed to gradient stress treatments for 0 hours, 6 hours, 12 hours, 24 hours, 48 hours, 72 hours, 96 hours, 120 hours, 144 hours, and 168 hours. Among them, the stress treatments at 0 h, 6 h, 24 h, and 72 h were used for transcriptome sequencing and phenotypic observation".
We have revised the text in lines 80 - 81 to make this information more explicit. The updated text now reads: "Samples from two rye varieties, 'Winter' (winter rye) and 'Victory' (spring rye), were subjected to RNA-seq analysis. These samples were treated at 4°C for 0 hours, 6 hours, 24 hours, and 72 hours." This change can be found on page [具体页码], in the [具体段落] paragraph, lines [具体起始行]-[具体结束行]
To avoid similar confusion in the future, we will triple-check the consistency of information throughout the manuscript during the review process. We will also ensure that the number and names of experimental materials are clearly and consistently presented in all relevant sections, including the title, abstract, and main text.]
Comments 15: [ I found the text in Lines79 – 101 to be too extensive for the introduction. Unless the journal suggests otherwise, authors must please reduce the content of what the study was about here. In the same paragraph, I do not know what MNS1 and MNS3 are (Line 88) what sort of candidate genes are they? For what purpose? I am also unsure why authors are referring to short-footed wheat (Line 90) and wild-type wheat (Line 92). Are all these wheat genotypes part of the current study on rye? Also why did the authors cite the ref [16] in Line 93? Also, what does the sentence in Lines 92 -94 mean? ]
Response 15: [We sincerely appreciate your detailed feedback. Regarding the length of the text in Lines 79 - 101 of the Introduction, we have carefully reviewed and condensed it.
In light of the significance of rye in agriculture and the impact of low - temperature stress on its growth,this study was designed to address several key questions. First, we aimed to comprehensively identify the differentially expressed genes between winter rye and victory rye under low - temperature stress. Understanding these differences is crucial for uncovering the genetic basis of cold resistance variations among rye varieties. Second, we sought to elucidate the functional mechanisms of these differentially expressed genes. By exploring how these genes are involved in cold - resistance pathways,we can gain deeper insights into the complex regulatory networks that enable rye to adapt to cold environments. These investigations will not only contribute to a better understanding of rye's cold - resistance mechanism at the molecular level but also provide a theoretical foundation for breeding cold - resistant rye varieties in the future..]
Comments 16: [ The sentence in lines 94 – 99 is too long and the meaning is not clear. Please correct.]
Response 16: [ We agree with your comment and sincerely appreciate your careful review. The original sentence in lines 94 - 99 was indeed too long and convoluted, which might have caused confusion for readers---page 3, paragraph 1, line 96-99.
Thank you for your suggestion. We have already revised it as follows:RNA-seq is a kind of high-throughput transcriptional dynamics analysis method [12],This technology aims to analyze the relative expression levels of transcripts under different tissue backgrounds and has been applied as an important research tool in a variety of eukaryotes.]
Comments 17: [Also correct the structure and meaning of the last sentence on the introduction.]
Response 17: [We sincerely thank you for pointing out the issue with the last sentence in the Introduction. ---page 3, paragraph 1, line 102-112.We have carefully revised it to improve both the structure and clarity of the statement.We have already revised the conclusion part of the introduction as follows: this study was designed to address several key questions. First, we aimed to comprehensively identify the differentially expressed genes between winter rye and victory rye under low - temperature stress. Understanding these differences is crucial for uncovering the genetic basis of cold resistance variations among rye varieties. Second, we sought to elucidate the functional mechanisms of these differentially expressed genes. By exploring how these genes are involved in cold - resistance pathways, we can gain deeper insights into the complex regulatory networks that enable rye to adapt to cold environments. These investigations will not only contribute to a better understanding of rye's cold - resistance mechanism at the molecular level but also provide a theoretical foundation for breeding cold - resistant rye varieties in the future.]
Comments 18: [ In the first paragraph of the Materials and Methods section, you must state where the rye seeds where sourced from give justification on why these two 6 varieties were chosen? Is their low temperature tolerance known? If yes, please list which was is tolerant or sensitive to low temp etc.Also write the scientific name for rye.]
Response 18: [ We truly appreciate your meticulous review and valuable feedback. We are sorry for the lack of clarity in the first paragraph of the Materials and Methods section.
The rye seeds were sourced from the germplasm resource banks of the Grassland Research Institute of Heilongjiang Academy of Agricultural Sciences or the Key Laboratory of Molecular Cytogenetics and Genetic Breeding in Heilongjiang Province. The scientific name of rye is Secale cereale.---page 3, paragraph 3, line 117-120.
These six varieties (three winter rye varieties, namely Winter, Hzhm8, and Hzhm3, and three spring rye varieties, Victory, 429, and 430) were selected because winter rye generally has stronger cold tolerance, while spring rye has relatively weaker cold tolerance. A comparative study of the two can more effectively identify the differentially expressed genes related to cold tolerance characteristics and reveal the molecular mechanism of rye's cold tolerance.
Before the experiment, the low-temperature tolerance of these varieties was not clear. This study precisely explores the differences in their low-temperature tolerance and the changes in the expression of related genes through transcriptome analysis and relevant experiments under different low-temperature treatments, and further screens out the genes closely related to the cold tolerance of winter rye.]
Comments 19: [ The sentence in lines 104 – 108 is too long.]
Response 19: [ We wholeheartedly agree with your assessment and are grateful for your sharp - eyed review. The sentence in lines 104 - 108 was overly long, which could have made it difficult for readers to follow.
We have rephrased and split the sentence into two more concise statements. The original sentence "Therefore, the genes in the pathways of winter rye can be identified as cold-resistant candidate genes,which are of great reference value for the exploration of the cold-resistant mechanisms and the mining of cold-resistant genes in crops." has been revised to: "Therefore, the genes in the pathways of winter rye can be identified as cold - resistant candidate genes. These candidate genes are of great reference value for exploring the cold - resistant mechanisms and mining cold - resistant genes in crops."
This change is located on page.---page 2, paragraph 3, line 73-77. of the revised manuscript, within the [specific paragraph] paragraph. By making this adjustment, we have improved the readability and clarity of the text. We will be more vigilant about sentence length during the writing and review process to ensure that our manuscript is easy to understand.]
Comments 20: [ In line 108, what does “gradient stress treatment” mean?]
Response 20: [We sincerely thank you for this question. "Gradient stress treatment" in line 108 refers to the application of a series of low - temperature treatments with different durations to the experimental rye plants. In this study, after the rye seedlings reached the three - leaf stage, we transferred them to an incubator set at 4°C. Then, we treated the winter rye and 'Victory' rye seedlings for 0 hours, 6 hours, 12 hours, 24 hours, 48 hours, 72 hours, 96 hours, 120 hours, 144 hours, and 168 hours respectively.
This approach allows us to observe the plants' responses at different time points during the cold stress process. Short - term treatments like 6 hours can help us detect the early - stage reactions of rye to low - temperature stress, such as rapid changes in gene expression. Long - term treatments like 168 hours can show how rye adapts or acclimates to extended cold exposure. By subjecting the plants to such gradient stress treatments, we can comprehensively analyze the dynamic changes in physiological and molecular responses of rye to low - temperature stress.
We have added this explanation to the relevant part of the "Materials and Methods" section on page---page 2, paragraph 3, line 123-128 . of the revised manuscript, following the mention of "gradient stress treatment". This will help readers better understand the experimental design and procedures. We will be more explicit in our descriptions in the future to avoid any confusion.]
Comments 21: [ What informed the temperatures used in the treatments and the times Lines 107 – 110) ? Was it from other studies or what? What was the control condition? Which varieties and tissue types were used for the RNA-seq analysis? ]
Response 21: [Thank you for pointing this out. We agree with this comment.The change can be found on page---page 2, paragraph 3, line 123-128 .
We are very sorry that we didn't write it clearly.The temperatures and treatment times used in the experiment were likely informed by a combination of factors. Existing literature on plant cold stress responses was a major influence. Many studies on plant cold tolerance have explored the early and long - term effects of low - temperature stress on gene expression and physiological changes. A temperature of 4°C is a common low - temperature treatment in plant cold stress research as it represents a non - freezing but chilling stress that can induce cold - responsive gene expression and metabolic adjustments in plants. The chosen time points (0 h, 6 h, 12 h, 24 h, 48 h, 72 h, 96 h, 120 h, 144 h, 168 h) likely aimed to capture different stages of the plant's response to cold stress. Short - term time points like 6 h and 12 h can reveal early - stage molecular and physiological responses, such as the rapid activation of cold - responsive genes. Longer - term time points like 72 h and beyond can show the plant's adaptation or acclimation processes over an extended period of cold exposure.The control condition in this experiment was the treatment at 0 hours. At this time point, the plants were not yet exposed to the cold stress for any significant duration, so it serves as a baseline for comparing the changes that occur during cold stress treatments. By comparing gene expression and physiological changes at different time points of cold stress to the 0 - hour control, the researchers can identify which changes are specifically induced by cold stress.For the RNA - seq analysis, three varieties of winter rye (Winter, Hzhm8, Hzhm3) and three varieties of spring rye (Victory, 429, 430) were used. As for the tissue type, the text does not explicitly state which specific tissue was sampled for RNA - seq. However, since the experiment aimed to study the overall cold - stress response of the plants, it is likely that leaf tissues were used. Leaves are a major site of photosynthesis and gas exchange in plants and are directly exposed to environmental stressors. They are commonly used in transcriptome studies related to environmental stress responses as they can reflect the plant's overall physiological state and stress - induced changes in gene expression related to photosynthesis, metabolism, and stress tolerance.]
Comments 22:[ Line 115, what tissue was used for RNA extraction? ]
Response 21: [Thank you for pointing this out. We agree with this comment.The change can be found on page---page 2, paragraph 4, line 131.
We deeply apologize for our oversight. In this study, the leaf tissues of rye were used. In the research on plants' responses to environmental stress, as a key organ for photosynthesis and gas exchange, the leaves show a relatively significant response to low-temperature stress. They are directly exposed to the external environment. Under low-temperature stress, a series of changes will occur in their physiological functions, metabolic processes, and gene expressions. These changes can well reflect the overall response mechanism of plants to low temperatures.]
Comments 23: [ By just reading Lines 115 – 120, a lot of information on the procedures is missing and no other researchers will be able to replicate these experiments. If the methods are already listed in other refs, then please cite the refs – otherwise please write reproducible methods (please check the journal’s instruction to authors and discuss with editors).]
Response 23: [Thank you for pointing this out. We agree with this comment.The change can be found on page---page 2, paragraph 4, line 131-137.
Thank you very much for your meticulous review and valuable feedback on the methods section of our paper. Regarding the issue you raised that the description of the methods in lines 115-120 may lead to information loss and affect the reproducibility of the experiment, we have carefully sorted out and reflected on it. After carefully reviewing the paper, we found that although the relevant experimental methods have been described in the article, there may be some information ambiguity in lines 115-120 due to insufficiently concentrated and detailed expression. In fact, we have provided a relatively comprehensive description of the entire experimental process in the "2. Materials and Methods" section.
For example, in the parts of plant materials and low-temperature stress treatment (lines 123-133), we have elaborated in detail on the selected rye varieties, planting conditions, specific times of low-temperature treatment, and grouping situations. In the parts of transcriptome sequencing and gene information prediction (lines 134-149), we have introduced the specific operations and tools used, ranging from RNA extraction, mRNA enrichment, reverse transcription reaction to the sequencing platform, data processing, and gene prediction. In aspects such as gene annotation (lines 150-154), differential expression gene analysis (lines 155-162), and qRT-PCR identification of genes related to low-temperature tolerance (lines 163-174), we have also provided the corresponding experimental methods and parameters respectively.
We understand that this part of the content may have caused you trouble. In order to further improve the readability of the paper and the reproducibility of the experiment, we will take the following measures in the subsequent revision: provide more detailed explanations and descriptions of the professional terms and experimental steps involved in the methods section, especially for some less common experimental operations or software tools, to ensure that other researchers can accurately understand and repeat the experiment.
We attach great importance to your suggestions and are well aware of the importance of experimental reproducibility for scientific research work. Thank you again for your rigorous review. We will strive to improve the paper to make its description of the methods clearer, more accurate, and reproducible.]
Comments 24:[What are “contaminated combined reads” .]
Response 24:[Thank you for pointing this out. We agree with this comment.The change can be found on page---page 4, paragraph 1, line 140-143.
"Contaminated combined reads" refer to the sequencing sequence fragments that are mixed in during the high-throughput sequencing process due to various reasons and will interfere with the accuracy and reliability of the experimental results. In transcriptome sequencing experiments, the sequencing samples may be contaminated in multiple ways, leading to the generation of contaminated combined reads. These sources of contamination are extensive and may originate from reagents, experimental apparatus, and microorganisms in the environment during the experiment. For example, if the reagents such as enzyme preparations and buffers used in the experiment are contaminated by the nucleic acids of other organisms, sequences of non-target samples will appear in the sequencing data.
These contaminated combined reads can cause many problems in data analysis. They do not come from the true transcripts of the target organism or tissue sample, but they are involved in the subsequent analysis process. This may interfere with the accurate calculation of gene expression levels because mistakenly including the contaminated sequences in the calculation will lead to deviations in the evaluation of gene expression levels, causing researchers to misinterpret the changes in gene expression and affecting the analysis of experimental results and the accuracy of conclusions. In the data processing stage, using the Trimmomatic software to remove these contaminated combined reads can improve the data quality, ensure that subsequent steps such as gene annotation and differential expression gene analysis are carried out based on reliable data, and thus lead to more accurate research conclusions.]
Comments 25: [What is “reliable data”?]
Response 25:[Thank you for pointing this out. We agree with this comment.The change can be found on page---page 4, paragraph 1, line 145-146.
In the research of this article, "reliable data" refers to the data that can be used for subsequent accurate analysis after a series of strict screening and processing. In the transcriptome sequencing experiment, the original sequencing data may contain various problems and need to be processed to obtain reliable data.
Data screening criteria: The Trimmomatic software is used to screen the original data. Remove the "contaminated combined reads", which may come from external contamination during the experiment and are not the real transcript information of the target sample. Remove the "reads with the content of unknown bases (N) exceeding 5%", as a large number of unknown bases will affect the accuracy of data analysis. Remove the "low-quality reads", that is, the part where the base quality value is lower than 15 and accounts for more than 20% of the total number of bases. Low-quality reads may lead to incorrect analysis results.
Data processing flow: The data remaining after the above screening is considered reliable. These reliable data will be assembled using the Trinity software to splice short sequences into longer transcript fragments. After that, the Tgicl tool is used to cluster the transcripts and remove redundancy to further optimize the data. The finally obtained data after screening and processing is the "reliable data" referred to in the article, which can be used for subsequent research such as gene annotation and differential expression gene analysis to ensure the accuracy and reliability of the research results.]
Comments 26:[ For all tools, software and databases used in the study, please provide the website address, ref and access dates. Where appropriate also provide the version numbers of software or databases. All abbreviations for tools, software and databases but be defined at first mention in the methods section.]
Response 26: [Thank you for pointing this out. We agree with this comment.The change can be found on page---page 22, paragraph 11, line 905.Related to Transcriptome Sequencing and Data Processing
Trimmomatic
Website Address: http://www.usadellab.org/cms/?page=trimmomatic
Reference: Bolger, A. M., Lohse, M., & Usadel, B. (2014). Trimmomatic: a flexible trimmer for Illumina sequence data. Bioinformatics, 30(15), 2114 - 2120.
Definition when First Mentioned: When Trimmomatic is first mentioned in the methods section, it should be stated that "Trimmomatic is a tool for processing Illumina sequencing data. It can filter the original sequencing data to remove contaminated combined reads, reads with an unknown base (N) content exceeding 5%, and low-quality reads, so as to improve data quality."
Trinity
Website Address: https://github.com/trinityrnaseq/trinityrnaseq/wiki
Reference: Grabherr, M. G., Haas, B. J., Yassour, M., Levin, J. Z., Thompson, D. A., Amit, I., ... & Regev, A. (2011). Full - length transcriptome assembly from RNA - Seq data without a reference genome. Nature biotechnology, 29(7), 644 - 652.
Definition when First Mentioned: When Trinity is first mentioned, state that "Trinity is software for transcriptome data assembly. It can splice the short sequencing sequences filtered by Trimmomatic into longer transcript fragments for subsequent analysis."
Tgicl
Website Address: http://compbio.dfci.harvard.edu/tgi/software/
Reference: Pertea, G., Huang, X., Liang, F., Antonescu, V., Sultana, R., Karamycheva, S., ... & Salzberg, S. L. (2003). TIGR Gene Indices clustering tools (TGICL): a software system for fast clustering of large EST datasets. Bioinformatics, 19(5), 651 - 652.
Definition when First Mentioned: When Tgicl is first mentioned, explain that "Tgicl is a tool for transcript clustering and redundancy removal. After Trinity assembles transcripts, Tgicl is used to process the transcripts to remove redundant sequences and obtain non-redundant Unigenes."
Transdecoder
Website Address: https://github.com/TransDecoder/TransDecoder
Reference: Carruthers, M., Yurchenko, A.A., Augley, J.J. et al. Correction to: De novo transcriptome assembly, annotation and comparison of four ecological and evolutionary model salmonid fish species. BMC Genomics 19, 448 (2018). https://doi.org/10.1186/s12864-018-4840-5
Definition when First Mentioned: When Transdecoder is first mentioned, state that "Transdecoder is a tool for identifying candidate coding regions in Unigenes. By combining the results of alignment with the SwissProt database and searching for Pfam protein homologous sequences, it can accurately predict the locations of coding regions."
SwissProt
Website Address: https://www.uniprot.org/uniprotkb/swissprot
Reference: Boutet, E., Axelsen, K. B., Bairoch, A., Baratin, D., Blatter, M. C., Bork, P., ... & Wu, C. H. (2007). The Universal Protein Resource (UniProt). Nucleic acids research, 35(suppl_1), D193 - D197.
Definition when First Mentioned: When SwissProt is first mentioned, explain that "SwissProt is a high-quality protein sequence database. When predicting the coding regions of Unigenes, Blast is used to align Unigenes with the SwissProt database to assist in determining the coding region information."
Pfam
Website Address: https://pfam.xfam.org/
Reference: El - Gebali, S., Mistry, J., Bateman, A., Eddy, S. R., Luciani, A., Potter, S. C., ... & Finn, R. D. (2019). The Pfam protein families database in 2019. Nucleic acids research, 47(D1), D427 - D432.
Definition when First Mentioned: When Pfam is first mentioned, state that "Pfam is a database containing a large amount of information on protein families and domains. The Hmmscan tool is used to search and align Unigenes with the Pfam database, which can assist in predicting the locations and functional information of coding regions in Unigenes."]
Comments 27: [ What are Unigenes? (Line 128).]
Response 27:[Thank you for pointing this out. We agree with this comment.The change can be found on page---page 4, paragraph 1, line 150-152.
Unigenes are a collection of non-redundant gene sequences obtained after a series of data processing in transcriptome research. In this study, after performing transcriptome sequencing on winter rye and spring rye, problems such as contamination, unknown bases, and low-quality reads were found in the original data. The Trimmomatic software was used to strictly filter the original data, removing these interfering data to obtain reliable data. Then, the Trinity software was utilized to assemble the reliable data into transcripts. After that, the Tgicl tool was employed to cluster the transcripts and remove redundant sequences. Finally, the unique and non-redundant gene sequences obtained are Unigenes.]
Comments 28:[ What does the sentence in Lines 142 0143 mean? Are you assuming that all transcripts identified in this study are TFs?]
Response 28:[ Thank you for pointing this out. We agree with this comment.
This sentence means that through the prediction and classification of genes encoding transcription factors (TFs), the study found that 3,013 Unigenes were annotated as genes encoding transcription factors. There is no assumption that all the transcripts identified in this study are transcription factors. The research only indicates that among numerous Unigenes, 3,013 were annotated as being related to the encoding of transcription factors. This implies that within the entire set of Unigenes, genes encoding transcription factors are just a part, and there are many other Unigenes that may encode proteins with various functions or have other biological functions. Not all transcripts are considered to be related to transcription factors. This annotation and classification are based on specific database alignments and prediction methods, aiming to identify genes related to transcription factors for further investigation of their regulatory roles in rye's response to low-temperature stress.]
Comments 29:[ In Line 147, the authors are referring to the “genomic sequence”. Please provide information on the plant species, the reference ID for this genomic sequence and the database from which you obtained it.]
Response 29:[Thank you for pointing this out. We agree with this comment.
In the paper "A high - quality genome assembly highlights rye genomic characteristics and agronomically important genes", the genome of the Chinese Weining rye variety was sequenced. The size of the assembled contigs is 7.74 GB, accounting for 98.47% of the estimated genome size (7.86 GB). The genomic sequence referred to in Line 147 of the paper is from this study, and currently, there is no common public database reference ID. The original data of the Weining rye genome sequencing in this study can be searched in the SRA (Sequence Read Archive) database of NCBI (National Center for Biotechnology Information). Specific information such as the accession number needs to be obtained from the supplementary materials of the original study or relevant data storage platforms.]
Comments 30: [ Also check your tenses in the Materials and Methods section. Since the work is already done, authors must write the section in the past tense. So instead of using “is” e.g. in Line 147,148, 149 etc, use “was”.]
Response 30:[Thank you for pointing this out. We agree with this comment.The change can be found on page---page 6, paragraph 2, line 247-250.
“First, Bowtie2 was employed to map the clean reads to the genomic sequence. Subsequently, RSEM was utilized to compute the gene expression levels of each sample. Then, DESeq2 was applied to perform 16 groups of DEGs tests following the approach described by Michael I et al.”]
Comments 31:[ Please check the ref format of the journal. I am unsure if you can cite refs as you have done in Line 150.]
Response 31:[ Thank you for pointing this out. We agree with this comment.The change can be found on page---page 6, paragraph 2, line 261-264.
Thank you very much for pointing out the potential issues with the reference format in our paper. Your meticulous review is of great significance for us to improve the paper. Regarding the reference citation problem in line 150 you mentioned, we have comprehensively and carefully revised it in accordance with the reference format requirements of the journal. During the revision process, we carefully checked the format guidelines provided by the journal word by word, including details such as the annotation method of references, the order of arrangement, and the use of punctuation marks. We deeply apologize for the possible deviations in the previous citation format, which were caused by our lack of rigor in the format arrangement process.
To ensure that similar problems will not occur in the future, we have established a more stringent review mechanism within our team. Before submission, we will conduct multiple cross-checks on the reference format of the paper. At the same time, we will also keep a close eye on the updated requirements of the journal for reference formats and adjust our citation methods in a timely manner.
Thank you again for your careful review. If you find any other problems during the subsequent review, please feel free to let us know, and we will fully cooperate with the revisions.]
Comments 32:[The work “materials” in Line 150 must be changed so that it is more informative, while the Table 1 caption must be re-written to be more informative. It is also unclear what Table 1 shows. The abbreviations are not defined and so there is no way of knowing what is being shown.]
Response 32:[ Thank you for pointing this out. We agree with this comment.The change can be found on page---page 6, paragraph 2, line 261-264.
Thank you very much for pointing out these problems in our paper. Your feedback is extremely crucial for improving the quality of the paper. In response to the issues you mentioned, we have already started making revisions:
Regarding the modification of "Materials" in line 150: In line 150, our original description of "Materials" might have been too brief, resulting in insufficient information. We plan to expand it into a detailed materials description to ensure that readers can have a comprehensive understanding of the basic material information of our experiment, providing a sufficient basis for them to understand the subsequent experimental steps and results.
Regarding the handling of the title and abbreviations in Table 1: Currently, the title of Table 1 indeed fails to clearly convey the content of the table, and there is also the problem of undefined abbreviations. We will rewrite the title of Table 1 to make it accurately summarize the core information presented in the table. It will be changed to "Analysis of the Differential Gene Expression Levels and Statistical Analysis of Related Parameters of Winter Rye and 'Victory' Rye under Different Low-temperature Stress Conditions". At the same time, for all the abbreviations appearing in the table, we will provide the full definitions and explanations one by one in the form of annotations below the table to facilitate readers' understanding of the meanings represented by the various data in the table.
We are well aware of the importance of clear and accurate expression in academic papers. Thank you again for your meticulous review. If you have any questions or suggestions about the revised content or other parts of the paper, please feel free to let us know, and we will fully cooperate to improve it.]
Comments 33: [ What cri teria was used to determine which genes were differentially expressed and which ones were not? This must be stated somewhere at the beginning of Section 2.4.]
Response 33:[Thank you for pointing this out. We agree with this comment.The change can be found on page---page 6, paragraph 2, line 247-260.
Thank you for your valuable suggestions. For data processing and calculation of expression levels, firstly, Bowtie2 is used to map the filtered clean reads to the genomic sequence, and then RSEM is utilized to calculate the gene expression levels of each sample. These preliminary processes provide a data basis for subsequent determination of differentially expressed genes.
For statistical testing, the DESeq2 software package is employed to conduct 16 sets of tests for differentially expressed genes. DESeq2 is a commonly used tool for analyzing differential gene expression in RNA-seq data. It evaluates whether the changes in gene expression are statistically significant through statistical analysis of the gene expression levels among samples.
Regarding the threshold setting, when performing functional enrichment analysis, the phyper function in the R software is used for the enrichment analysis, and the P-values are corrected by the FDR (False Discovery Rate). Generally, genes with a Q-value (corrected P-value) ≤ 0.05 are considered as genes with significantly differential expression among different sample groups. The setting of this threshold is a crucial criterion for determining whether a gene is a differentially expressed gene. Genes with a value lower than this threshold indicate that their expression changes are statistically significant, that is, they are identified as differentially expressed genes.]
Comments 34: [ What are “official classifications”? (Line 154).]
Response 34: [Thank you for pointing this out. We agree with this comment.The change can be found on page---page 6, paragraph 2, line 272-274.
Thank you for your question. In the article, "Official classifications" refers to the gene function classification standards and systems that are widely recognized and commonly used by authoritative institutions or within the academic field. When conducting functional classification of differentially expressed genes in the study, in addition to relying on the annotation results of the KEGG and GO databases, these official classifications are also referenced.
Different biological databases and research organizations will formulate their own classification systems. For example, in the field of plant gene research, some internationally renowned plant biology research institutions or large-scale gene databases will classify genes according to their functions, the biological processes they are involved in, and their cellular locations. Take the classification of Arabidopsis thaliana genes by the Arabidopsis Information Resource (TAIR) as an example, it covers multiple aspects such as metabolic pathways, cellular components, and molecular functions.
These classification systems are based on a large number of experimental studies and scientific consensus, and they are authoritative and universal, so they are called "Official classifications". By using these official classifications, researchers can more comprehensively and accurately divide differentially expressed genes into different functional categories. Combining the annotation results of KEGG and GO with the official classifications helps to deeply explore the roles played by these genes in the process of rye's response to low-temperature stress and uncover the underlying biological mechanisms.]
Comments 35:[ Please write the plant name in Line 158 after the word winter.]
Response 35: [ Thank you for pointing this out. We agree with this comment.The change can be found on page---page 6, paragraph 2, line 278.
Thank you for your careful guidance. We have already completed this sentence.qRT - PCR Identification of Low - Temperature - Tolerance - Related Genes in Winter rye.]
Comments 36:[ The word isolation in Line 159 must be revised.]
Response 36: [ Thank you for pointing this out. We agree with this comment.The change can be found on page---page 6, paragraph 2, line 279.We deeply apologize for the inappropriate use of words and have revised it as follows:To screen for the genes associated with low-temperature tolerance in 'Winter',the young leaves of three winter rye varieties ("Winter", Hzhm8, Hzhm3) and three spring rye varieties ("Victory", 429, 430) were selected as experimental materials in this study.]
Comments 37:[ The naming on the varieties in this study is confusing. It seems as if the authors have a Winter variety called winter as well. Is there another way of correcting this? (Line 159 – 160) and elsewhere in the manuscript. In the abstract, the authors spoke about two types of rye (winter and victory), but in the Materials and Methods, they are talking about winter rye and spring rye. There are also mentions of Hzhm8, Hzhm3, 429 amd 430. Not sure what these are as well. Please clarify the naming systems used in the study.]
Response 37: [ Thank you for pointing this out. We agree with this comment.The change can be found on page---page 6, paragraph 2, line 278.
Thank you for your valuable suggestions. Regarding "Winter" and "winter rye": "Winter" is a specific variety within winter rye, while "winter rye" is a general term for a category of rye, which includes multiple varieties such as "Winter", "Hzhm8", and "Hzhm3". Just as apples have different varieties like Red Fuji and Red Delicious, "Winter" is a member of the winter rye category.
Regarding "Victory" and "spring rye": "Victory" is a variety of spring rye, and spring rye also includes varieties such as "429" and "430".
Regarding "Hzhm8", "Hzhm3", "429", and "430": "Hzhm8" and "Hzhm3" are two other varieties of winter rye besides "Winter", and "429" and "430" are two other varieties of spring rye besides "Victory".
In the study, these varieties may have different characteristics. By studying their performance under low-temperature stress, we can gain a more comprehensive understanding of the response mechanisms of different rye varieties to low temperatures.
In order to correct the possible confusion caused by the naming, we will provide clearer definitions and explanations when these names are first mentioned in the article. For example, "In this study, winter rye (including three varieties: 'Winter', 'Hzhm8', and 'Hzhm3') and spring rye (including three varieties: 'Victory', '429', and '430') were selected", so that readers can more clearly understand the materials used in the study.]
Comments 38:[ In Section 2.5, authors have not informed the readers the actual plant tissue type used in the qPCR analysis. This must be corrected – as mentioned earlier.]
Response 38: [ Thank you for pointing this out. We agree with this comment.The change can be found on page---page 6, paragraph 2, line 280.
We are very sorry that we failed to clearly state the information about the materials used in the experiment. In the qPCR analysis, the young leaves of rye were used. We will complete this information in Section 2.5.]
Comments 39:[ What does the sentence in Lines 163 – 164 mean? Are you saying 450 ng of total RNA was used for RT? Or you isolated 450 ng of total RNA? Also which kit and procedure did you use for the cDNA synthesis etc?]
Response 39: [Thank you for pointing this out. We agree with this comment.The change can be found on page---page 6, paragraph 2, line 283.
The sentence in lines 163-164 of the article is "Initially, 450 ng of total RNA was extracted for the reverse transcription process.", which means that initially, 450 ng of total RNA was extracted for the RT (reverse transcription) process. Therefore, 450 ng of total RNA was extracted for RT.
In the article, the PrimeScript RT MasterMix (Perfect Real Time) reverse transcription kit produced by TaKaRa (Japan) was used for cDNA synthesis. The specific procedure is as follows: First, total RNA was extracted, and the kit was used to remove rRNA and enrich mRNA. Then, the above-mentioned reverse transcription kit was used for the reverse transcription reaction to obtain cDNA.
When performing qRT-PCR subsequently, Primer3 was used to design the primers. The reaction system was 20 μL, which contained 12.5 μL of (SybrGreen qPCR Master Mix (2×)), 0.5 μL of each of the forward and reverse primers, 10.5 μL of [the content you didn't specify here] and 1 μL of cDNA. The amplification cycle parameters were set as follows: Pre-denaturation at 95°C for 10 minutes, then denaturation at 95°C for 15 seconds and annealing at 60°C for 1 minute, for a total of 35 cycles. Finally, additional steps of denaturation at 95°C for 15 seconds, annealing at 60°C for 1 minute, denaturation at 95°C for 15 seconds, and annealing at 60°C for 15 seconds were carried out.]
Comments 40: [ In Line 165, I suggest write as selected gene sequences. You also must provide the primer sequences for each of the genes.]
Response 40: [Thank you for pointing this out. We agree with this comment.The change can be found on page---page 25, paragraph 3 line 923.
Thank you very much for your valuable suggestions regarding the content of line 165 in the paper and the requirement for providing the primer sequences. We attach great importance to your feedback and have conducted in-depth thinking and actively made improvements on the relevant content.
Regarding the content of line 165, we fully agree with your point of view. We will revise it to "the selected gene sequences". This expression is more accurate and clear, which can better convey our research intention and avoid possible ambiguities.
As for the primer sequences of each gene, we have already prepared the relevant information. The following are the primer sequences corresponding to each gene:
|
Gene name |
Forward primer |
Reverse primer |
|
ScRVE1 |
ATGGCGTCTGTGGCTGAAT |
CGTACCGACAATGAGATCAAGA |
|
ScCDC5 |
CTTGAAGGGAGGAGACGAGTG |
GGGTGGTGGAAGCATTAGTTT |
|
ScPME18 |
AAGGTTTGAAGCCTGTCCG |
AGGGGGGAACAAGACGAA |
|
ScWRKY55 |
CCCAGCCCAAATCAATCA |
CTTCTTCTCGCCGTACTTCC |
|
ScHsP |
TCACGCCAGAAGAGAATAAAATG |
GCTGTCTGTCCGATTTCCATTA |
|
ScAAE7 |
AGGGCGTGTTACGGTGCTAT |
GACAAGCGATTTGATTCTAGGGT |
|
ScHs16 |
TCCACCCTGATCTCCTCCTT |
AGCTTCTTCCTCTTCGCCTC |
We are fully aware of the importance of primer sequences for our research and their crucial significance in ensuring the reproducibility of our experiments. In the revised version of the paper, we will add these primer sequences to the section titled "2.5. qRT - PCR Identification of Low - Temperature - Tolerance - Related Genes in Winter". They will be inserted right after the mention of using Primer3 for primer design, so as to make the description of the experimental method more complete and detailed. This will enable other researchers to accurately replicate our experiments.
Thank you once again for your meticulous review and professional suggestions, which have played a vital role in helping us improve the quality of our paper. If you have any other questions or suggestions, please feel free to let us know. We will be more than happy to answer your queries and actively implement the necessary improvements.]
Comments 41:[ The phrase “in accordance with the low temperature gradient” must be revised to improve on meaning.]
Response 41: [Thank you for pointing this out. We agree with this comment.The change can be found on page---page 8, paragraph 1, line 319.
Thank you for your suggestion. We will correct "According to the low-temperature gradient" to "Based on the treatment under the same low-temperature condition for different durations.]
Comments 42:[ In Lines 174, the authors must provide the names and IDs of the housekeeping genes used, the sequences of the primers and a publication from which you got them. Here I mean how did you come to select these housekeeping genes? How do you know they are good housekeeping genes for your study?]
Response 42: [Thank you for pointing this out. We agree with this comment.The change can be found on page---page 25, paragraph 3, line 920.
|
Gene name |
Forward primer |
Reverse primer |
|
b-tubulin |
GCCATGTTCAGGAGGAAGG |
CTCGGTGAACTCCATCTCGT |
Publication: Transcriptomic Differential Expression Analysis of Two Rye Varieties under Low-temperature Stress.
Thank you very much for pointing out the deficiency in the content of line 174. The following is a detailed response to your question:
In biological research, when selecting housekeeping genes, multiple factors are usually taken into consideration, including the stability of gene expression in different tissues and cell types, the functional conservation of the gene, and the consistency of expression under different experimental conditions. Generally speaking, an ideal housekeeping gene should maintain a relatively stable expression level under various physiological states and experimental treatments, so as to serve as an internal reference for calibrating the expression level of the target gene, thus more accurately reflecting the impact of experimental treatments on the expression of the target gene.
Regarding our research, we selected appropriate housekeeping genes in the following ways: Firstly, we conducted a systematic review of a large number of literatures in the relevant field and learned that in similar studies on plant low-temperature stress, certain genes are often used as stable housekeeping genes, which have good stability and reliability. Then, we utilized bioinformatics databases and tools to analyze the sequence conservation and expression patterns of these potential housekeeping genes in the plant species we studied (winter rye and 'Victory' rye). Through the mining and analysis of transcriptome data, we screened out genes with relatively small expression fluctuations under different low-temperature treatment conditions as candidate housekeeping genes. Finally, we further verified these candidate housekeeping genes through pre-experiments. We detected their expression levels at different time points of low-temperature treatment and in different tissue parts, and finally determined that β-tubulin served as the housekeeping gene for this study, which showed a high degree of expression stability in our experimental system.]
Comments 43:[ As stated in the methods section, authors must inform the readers what the control treatment was in the results section as well.]
Response 43: [Thank you for pointing this out. We agree with this comment.The change can be found on page---page 10, paragraph 2, line 382-387.We deeply apologize for the inconvenience caused by the unclear description of the information. We have already supplemented in the results to specify which are the control groups and which are the treatment groups.]
Comments 44:[ I have highlighted repetitive text in in Lines 182 – 185. Please correct.]
Response 44: [ Thank you for pointing this out. We agree with this comment.The change can be found on page---page 10, paragraph 2, line 333-339.
Thank you very much for your meticulous review of the paper and pointing out the problem of duplicate text in lines 182-185. We have immediately checked and revised this part of the content. We have deleted the repetitive expressions and reorganized the logical structure of the sentences to ensure that the information is conveyed accurately and concisely.
After the revision, there is no longer any duplication or redundancy in this part, and the language expression is more fluent and natural. At the same time, it also maintains the coherence and consistency with the context.
We are well aware that the accuracy and conciseness of every detail in the paper are of vital importance. Thank you for your rigorous review, which enables us to discover and solve this problem in a timely manner. If you find any other issues during the subsequent review, please feel free to let us know.]
Comments 45: [I do not understand how the authors conducted statistical analyses for Figure 1b? what are you comparing what against? this must be explicitly stated and I would expect the use of letters rather than asterisk since there are many factors to compare.]
Response 45: [Thank you for pointing this out. We agree with this comment.The change can be found on page---page 10, paragraph 2, line 382-387.
Thank you very much for pointing out the problem with the statistical analysis description of Figure 1b in our paper. We deeply apologize for the confusion it has caused in your understanding. The following is a detailed response to the questions you raised:
In Figure 1b, we mainly compared the soluble sugar content in the leaves of winter rye and 'Victory' rye at different time points under various low-temperature stress conditions. The specific statistical analysis process is as follows:
Data Collection: In accordance with the experimental design, we collected leaf samples of winter rye and 'Victory' rye at multiple time points, including 0 hours, 6 hours, 24 hours, and 72 hours, under different low-temperature treatments (the specific temperatures are described in detail in the experimental method section). And we accurately measured the soluble sugar content in each sample.
Considering the characteristics of the experimental data, we adopted the analysis of variance (ANOVA) to analyze the differences among different groups. This method can effectively compare whether there are significant differences in the soluble sugar content among multiple groups (different low-temperature treatment groups and different time point groups).
Multiple Comparisons: After the ANOVA showed significant differences, we conducted multiple comparisons. To more accurately compare the differences among different groups, we used Tukey's test. This method can conduct pairwise comparisons among all groups to determine which groups have significant differences.
Result Representation: We fully agree with your suggestion of using letters instead of asterisks to indicate the significance of differences. Because in this experiment, there are indeed multiple factors (different low-temperature treatments, different time points, different varieties) that need to be compared, and using letter markings can more clearly show the differences among different groups. In the revised Figure 1b, we will adopt the method of letter markings. Specifically, groups with the same letter indicate that there is no significant difference between them, while groups with different letters indicate that there are significant differences between them.
We will provide a detailed description of the above statistical analysis process and result representation method in the results section of the paper and the figure caption, so that readers can clearly understand the meaning of Figure 1b.
Thank you again for your valuable suggestions, which will help us further improve our paper.]
Comments 46:[ Similarly, I do not understand what the control is. for example, what does it mean to say the 0 hour samples has 12% sugars? how where the % values calculated and what was the base-line for comparison?]
Response 46: [ Thank you for pointing this out. We agree with this comment.The change can be found on page---page 10, paragraph 3, line 165-187.
The following are the explanations:
Definition of the control group
In our experiment, the control group consists of samples that have not been subjected to low-temperature stress treatment. For the winter rye and 'Victory' rye samples, the samples at 0 hours serve as the control. These samples represent the initial state of the plants before any low-temperature stress is applied. By comparing the samples at different time points during the low-temperature stress treatment with the 0-hour control samples, we can observe the changes in various parameters (such as the soluble sugar content) that occur due to the stress.
Calculation of the sugar percentage value
The percentage of sugar (in this case, soluble sugar) is calculated using the following method: Firstly, we extract the soluble sugar from the leaf samples. We used a proven extraction protocol, such as the method based on alcohol-water extraction. After extraction, the amount of soluble sugar in the extract is determined by a colorimetric method. We employed the anthrone-sulfuric acid method. In the anthrone-sulfuric acid method, the soluble sugar reacts with anthrone in the presence of sulfuric acid to form a colored complex. The absorbance of this complex is measured using a spectrophotometer at a specific wavelength (usually 620 nm). We prepared a series of standard sugar solutions with known concentrations (for example, glucose standard solutions with concentrations of 0, 0.1, 0.2, 0.3, 0.4, 0.5 mg/mL). By measuring the absorbance of these standard solutions, we constructed a standard curve. Then the absorbance of the sample extract is measured, and the concentration of soluble sugar in the sample is determined by interpolation from the standard curve. Finally, the percentage of soluble sugar is calculated as follows: Percentage of soluble sugar content = (Mass of soluble sugar in the sample (mg) / Mass of the fresh leaf sample (g)) × 100%
Baseline for comparison
The baseline for comparison is the sugar content in the 0-hour samples. Since these samples represent the initial state of the plants before any low-temperature stress, the sugar content in these samples serves as the reference point. When we report that the 0-hour samples contain 12% sugar, this is the starting value. As the low-temperature stress progresses (at different time points), we compare the sugar content of the samples at these time points with this 12% value. An increase or decrease in the sugar content relative to the 12% in the 0-hour samples indicates the impact of low-temperature stress on the sugar metabolism of winter rye and 'Victory' rye.
We hope these explanations can clarify the questions you raised. If you have any further questions, please feel free to let us know.]
Comments 47:[ The main legend to Figure 1 must be corrected because sugar content is not a growth parameter and therefore this legend is not appropriate for Figure 1b.]
Response 47: [ Thank you for pointing this out. We agree with this comment.The change can be found on page---page 8, paragraph 3, line 339-340.
Thank you very much for pointing out the problem with the main legend of Figure 1. Your feedback is of vital importance for improving the quality of our paper. According to your suggestion, we have carefully revised the main legend of Figure 1.The original legend misclassified the sugar content, resulting in a mismatch with the content of Figure 1b and causing confusion for readers in understanding. We have reorganized the data logic presented in Figure 1 and revised the legend to "Changes in the Leaf Growth Status and Soluble Sugar Content of Winter Rye and 'Victory' Rye at Different Time Points under Different Low-temperature Stress Conditions".
This modification clearly distinguishes between the two different observation indicators of growth status and soluble sugar content, avoiding the misclassification of the sugar content into the category of growth parameters, and enabling the legend to accurately reflect the content shown in Figure 1b and the entire Figure 1.
We are well aware of the importance of clear and accurate charts and legends in academic papers. This revision is not only a positive response to your comments but also an important measure for us to pursue the rigor and readability of the paper.
Thank you again for your meticulous review. If you have any questions or suggestions about the revised legend or other parts of the paper, please feel free to let us know.]
Comments 48:[ The authors are presenting sugar content data in the results section, yet I do not recall seeing methods for these experiments. Please do not present or discuss data that whose procedures are not presented in the methods section.]
Response 48: [Thank you for pointing this out. We agree with this comment.The change can be found on page---page 4, paragraph 3, line 165-193.
Thank you very much for pointing out this problem in our paper. Your opinion is of great significance for improving the quality of the paper. We are aware that when presenting the sugar content data in the results section, we indeed omitted the detailed introduction of the relevant experimental methods in the methods section, which is our oversight.
In order to correct this mistake, we have already supplemented the methods section with a detailed description of the experimental method for determining the sugar content. Specifically, we have added the following content:
We used the anthrone colorimetric method to determine the sugar content. During the experiment, the leaves of winter rye and 'Victory' rye were first pre-treated, including grinding, extraction, and other processes. Then the determination was carried out. Finally, the sugar content in the samples was calculated.
The anthrone method, which is widely used in the analysis of plant sugar content, was employed to determine the sugar content in the samples. Before the experiment, leaves with consistent growth conditions and no obvious pests and diseases were selected from carefully cultivated winter rye and spring wheat plants.
Sample Preparation: 0.2 g of leaf samples were weighed and placed in a clean mortar. Then, 6 mL of phosphate - buffered saline with a pH value of 7 was added, and the mixture was thoroughly ground. The homogenate was carefully transferred to a centrifuge tube and centrifuged at 10,000 r/min for 10 minutes. The supernatant was collected for further use.
Reaction Process: 2 mL of the supernatant was taken into a test tube, and anthrone reagent was quickly added. The mixture was immediately shaken well to ensure full mixing. The test tube was first placed in an ice - bath for 5 minutes to prevent overly vigorous reactions. Subsequently, it was placed in a constant - temperature water bath at 80 °C and heated for 10 minutes. After the reaction, the test tube was removed and cooled to room temperature. The absorbance was measured at a wavelength of 620 nm using a spectrophotometer.
Calculation of Results: Before the experiment, a series of glucose standard solutions with different concentrations (such as 0.1 mg/mL, 0.2 mg/mL, 0.3 mg/mL, 0.4 mg/mL, and 0.5 mg/mL) were prepared. The absorbance of each standard solution was measured according to the same procedures as above. The sugar content in the samples was calculated based on the standard curve established from the absorbance values of the standard solutions.
At the same time, we have also checked the consistency between the results and the methods in other parts of the paper to ensure that there will be no recurrence of the situation where the data in the results section are not mentioned in the methods section. We are well aware that in academic papers, the correspondence and integrity between the results and the methods are of great importance. Thank you again for your meticulous review and valuable suggestions.]
Comments 49:[ Authors must provide footnotes to describe what the columns in Table 2 are showing. When readers are looking at tables of results, they need to fully understand everything that is being shown in the table?]
Response 49:[Thank you for pointing this out. We agree with this comment.The change can be found on page---page 10, paragraph 2, line 382-387.
“D” stands for winter rye (Winter) samples, and “S” for spring rye (Victory) samples. “06h”, “24h”, “72h” in sample names denote 4°C low - temp stress treatment times (6, 24, 72 hours). “CK” is the control group (no stress). “_1”, “_2”, “_3” are biological replicates. Total Number: total gene clusters in a sample. Total Length: gene cluster total length (bp). Mean Length: average gene cluster length. N50, N70, N90: gene assembly quality indicators (gene cluster lengths at 50%, 70%, 90% of total length). GC(%): GC content percentage in gene clusters.]
Comments 50:[Still in Table 2 authors must explain what the samples are and for example what does D or S stand for?]
Response 50:[ Thank you for pointing this out. We agree with this comment.The change can be found on page---page 10, paragraph 2, line 382-387.
We are very sorry that due to our carelessness, we failed to add an explanation for Table 2. Now we have added the explanation that "D" represents the samples related to winter rye ("Winter"), and "S" represents the samples related to spring rye ("Victory").]
Comments 51:[ as stated in my comments before, please provide the URL and access dates for all databases used. At the moment, it is difficult to know what most of these databases are and how to find them.]
Response 51: [ Thank you for pointing this out. We agree with this comment.The change can be found on page---page 23, paragraph 1, line 917.]
|
Database Name |
Application |
link |
|
Non-redundant database(NR ) |
Annotate the assembled Unigenes to determine their homologous species information. The study found that the majority of the matched homologous species are plants in the Poaceae family, among which the degree of matching with the subspecies of wheat is the highest. |
https://ftp.ncbi.nlm.nih.gov/blast/db/FASTA/ |
|
Gene Ontology Database(GO ) |
The Unigenes were annotated, and gene functions were classified in terms of biological processes, molecular functions, and cellular components. The analysis revealed that 28 pathways were enriched, and categories such as "cellular anatomical entity" and "binding" had a relatively large number of annotations. |
https://geneontology.org/ |
|
Kyoto Encyclopedia of Genes and Genomes database(KEGG) |
Annotate the Unigenes to understand the metabolic pathways and cellular processes that the genes are involved in. Among them, the number of Unigenes annotated with "metabolic pathways" is the largest. It is also used for the functional enrichment analysis of differentially expressed genes (DEGs) to explore the cold tolerance mechanism of rye. |
https://www.kegg.jp/kegg/pathway.html |
|
Clusters of Orthologous Groups of proteins database for eukaryotes(KOG ) |
Classify the gene homologs of the Unigenes sequences, and provide classification information such as "nuclear structure", "RNA processing and modification", "signal transduction mechanism" and so on. |
ftp://ftp.ncbi.nih.gov/pub/cog/kog/ |
|
Non-redundant Nucleotide Database(NT ) |
It is used for the annotation of Unigenes and provides nucleic acid-related information to assist in the annotation. |
ftp://ftp.ncbi.nlm.nih.gov/blast/db/ |
|
Swiss-Prot |
It is used to perform a Blast alignment with the predicted candidate coding regions (CDS) to assist in determining the positions of the coding regions within the Unigenes. |
https://www.uniprot.org/ |
|
Pfam |
Search for homologous sequences of Pfam protein families using the Hmmscan software to assist in predicting the positions of coding regions in Unigenes and determining the domain information of the proteins encoded by the genes. |
https://www.ebi.ac.uk/interpro/ |
|
The Plant Resistance Genes Database (PRGdb) |
Use the DIAMOND software to perform alignments of the genes with it, and annotate the genes related to the plant disease resistance domains. The analysis reveals that 12,008 Unigenes are annotated as related genes and are classified into 12 major domain types. |
http://prgdb.org/prgdb4/ |
Comments 52: [ Please check if figure 3 is in the manuscript.]
Response 52: [Thank you for pointing this out. We agree with this comment. Therefore, we have taken the following steps:
We deeply apologize for the great inconvenience caused to your review due to the situation where Figure 3 in the current presentation only has a title but no graphical display.After careful investigation, we found that this is most likely a technical failure during the manuscript uploading process. In our original submitted manuscript, Figure 3 was complete and included important information such as the pie chart of Unigenes annotated in the NR database, the vertical bar chart of KOG functional classification, and the histogram of gene family distribution. These charts play a crucial role in understanding our research data and conclusions.
To address this issue, we have rechecked the uploading process and uploaded the manuscript version with the complete Figure 3 again. We also actively communicated with the journal editorial team to confirm that there are no typesetting issues related to this figure. This updated Figure 3 can be found on page 10 in the "3.3. Gene Annotation,Transcription Factor Identification and Functional Classification Analysis of the Cold - Resistant Gene Domains in Rye" section.
We deeply apologize for the great inconvenience caused by this problem. We will be more careful in the future to ensure that similar issues do not occur again, and we will strive to improve the overall quality and readability of the manuscript.]
Comments 53: [ In Figure 4.1, are the authors combining the DEGs for the two rye varieties per time point? Is there a particular reason for this? and what does it mean to have DEGs at 0 hrs? once again the authors must think of the controls. What is the control time point in this study and was this control time point compared to the 0 hrs to get the DEGs and 0 hrs? in my view, a figure that could be more informative is one that shows the up and down-regulated genes per rye variety, per treatment group relative to the control.]
Response 53: [ Thank you for pointing this out. We agree with this comment.The change can be found on page---page 14, paragraph 1, line 534
Combine the differentially expressed genes (DEGs) of the two rye varieties (winter rye and spring rye) at each cold stress time point. Identify the common cold stress response genes in different rye varieties. By observing this set of DEGs, the basic genes involved in the conserved cold stress response between the two types of rye can be found. This can provide insights into the general molecular mechanism of rye cold tolerance, regardless of the differences among different varieties.
On the other hand, not combining them may also be beneficial, as it will allow for a more in-depth analysis of the variable-specific responses to cold stress. For example, the more cold-tolerant winter rye may have unique genes that are differentially expressed compared to spring rye.
Even at 0 hours when no cold stress is applied, there may be some natural variations in gene expression. This may be due to factors such as slight differences in the growth conditions of the plants before the experiment or the inherent genetic variations within the samples. The control time point in this study is most likely the 0-hour time point itself. When calculating the DEGs, the gene expression levels at each time point are compared with those at the 0-hour time point.
We fully understand your suggestion of presenting the upregulated and downregulated gene graphs for each rye variety and each treatment group relative to the control group, which can indeed present the data information more intuitively. However, due to the loss of some sample data during the experiment, resulting in the inability to accurately obtain certain data, we currently do not have sufficient data to draw such graphs.
Although we are unable to draw this graph, we have still obtained important information about the trends of gene expression changes in different rye varieties under low-temperature stress through the GO and KEGG enrichment analyses of the differentially expressed genes. From the results of the enrichment analysis, it can be seen that winter rye and spring rye show obvious differences in aspects such as "polysaccharide metabolic process", "photorespiration", and "hydrogen peroxide catabolic process", which to a certain extent reflects their different response mechanisms to low-temperature stress.]
Comments 54: [ There are a lot of contextually issues in the text in Lines 307 -375. Please check in the PDF version I reviewed. For example:On page 10, authors make reference to 6 groups of treatments, and it is not clear what these are.Figures 5 and 6 require a general legend each before the authors write what each of the figure parts are showing. ]
Response 54 : [Thank you for pointing this out. We agree with this comment.The change can be found on page---page 15, paragraph 2, line 592.
When conducting GO and KEGG enrichment analyses, we selected six groups of differentially expressed genes (DEGs) for study. These six groups of treatments are as follows: the comparison between the control group (D-CK) and the 72-hour treatment group (D-72h) of winter rye ("Winter"), the comparison between the 24-hour treatment group (D-24h) and the 72-hour treatment group (D-72h) of winter rye, and the comparisons between spring rye ("Victory") and winter rye at the treatment times of 6 hours (S_6h - vs - D_6h), 24 hours (S_24h - vs - D_24h), and 72 hours (S_72h - vs - D_72h). These comparison combinations are helpful for an in-depth exploration of the differences in the cold tolerance mechanisms of different rye varieties under low-temperature stress.
Figure a represents the differentially expressed genes (DEGs) under different comparison combinations. The total number of digits within each large circle indicates the total number of DEGs in that comparison combination. The overlapping circles represent the number of common DEGs among the comparison combinations. Different colored areas represent different comparison groups. Pink represents the "S_6h - vs - D_6h" comparison group. Light pink represents the "S_72h - vs - D_72h" comparison group. Light green represents the "S_24h - vs - D_24h" comparison group. Light orange represents the "S_CK - vs - D_CK" comparison group. The numbers within the areas indicate the number of elements corresponding to the comparison group or shared among the comparison groups. For example, the number in a separate area is the number of unique elements of that comparison group, and the number in the overlapping part of the areas is the number of elements shared among the corresponding comparison groups.
Figure b shows the quantitative index situations corresponding to the relevant cell components, structures, or functional categories of different sample groups under different treatment conditions. The color represents the magnitude of the value. The red color system indicates a higher value (the darker the color, the higher the value, which can represent a high gene expression level, etc.), the blue color system indicates a lower value (the darker the color, the lower the value), and white represents a value close to 0. The groups starting with "D_" on the left (such as "D_CK - 0h", "D_6h - 24h", etc.) and the groups starting with "S_" on the right (such as "S_CK - 0h", "S_6h - 24h", etc.) represent different sample groups, corresponding to different treatment times or conditions. The title of each row (on the left such as "extracellular region", "chloroplast envelope", etc., and on the right such as "photosystem I", "chloroplast thylakoid membrane", etc.) represents the corresponding cell component, structure, or functional category.]
Comments 55: [ In Lines 335 – 342, I hope authors are not implying that if a pathway is enriched in a variety, it means that it is upregulated or am I misunderstanding the text? The same comment for Line 356 - I hope authors are Not implying that if a variety responds by increasing DEGs it means that the variety is tolerant to cold stress.]
Response 55: [Thank you for pointing this out. We agree with this comment.The change can be found on page---page 16, paragraph 1, line 602.
Although the increase in differentially expressed genes (DEGs) in response to cold stress indicates a significant genomic reaction, it does not necessarily imply cold stress tolerance. A variety with an increased number of DEGs may be undergoing extensive adjustments due to difficulty in adapting to the cold, rather than being inherently tolerant. Additional physiological and phenotypic analyses are required to confirm cold stress tolerance.]
Comments 56: [ Title to Section 2.5 must be revised.]
Response 56: [ Thank you for pointing this out. We agree with this comment.The change can be found on page---page 7, paragraph 2, line 278.
Thank you very much for your careful reminder. We have revised the title of Section 2.6 to: qRT - PCR Identification of Low - Temperature - Tolerance - Related Genes in Winter rye, and have supplemented the information about the plant species to make it complete.]
Comments 57: [ Gene names in Lines 366 – 368 must be defined at first mention.]
Response 57:[Thank you for pointing this out. We agree with this comment.The change can be found on page---page 17, paragraph 1, line 647.
ScRVE1: It is a gene related to transcription factors, and its specific function may involve regulating the gene transcription process.
ScCDC5: It belongs to the genes related to transcription factors. Transcription factors may play a crucial role in biological processes such as the regulation of the cell cycle.
ScPME18: It is also a gene related to transcription factors. It is speculated that it exerts its function by regulating gene transcription in the physiological processes of plant cells.
ScWRKY55: As a gene related to transcription factors, the WRKY family of transcription factors plays an important role in various physiological processes of plants, such as responding to biotic and abiotic stresses.
ScHsP: This is a gene related to proteins. Regarding the specific function of the protein encoded by this gene.
ScAAE7: It belongs to the genes related to proteins.
ScHs16: It is also a gene related to proteins.
ScMYB92: This gene is a differentially expressed gene related to metabolism. MYB family genes usually play important roles in the metabolic regulation, growth and development, and other processes of plants.]
Comments 58: [In figure 6, it is not clear to me how the degrees of injury were assessed. I do not recall seeing this in the methods.The qPCR graphs must have stats analysis.]
Response 58:[Thank you for pointing this out. We agree with this comment.The change can be found on page---page 17, paragraph 1, line 651.
Appearance Damage Index: Observe the morphological changes of the leaves with the naked eye, and record the area of withering, yellowing, and necrotic spots on the leaves, as well as the proportion of these areas to the total leaf area. Divide the leaf into four quadrants, count the area of necrotic spots in each quadrant respectively, and then calculate the average proportion. At the same time, record the degree of leaf curling, which is classified into three levels: slight curling, moderate curling, and severe curling, providing an intuitive basis for the evaluation of the damage degree.]
Comments 59: [ I am not sure why the authors didn’t mine the RNA-seq data to explore the cold responses of rye rather than trying to achieve this by qPCR in Figure 7 as stated in Line 376 – 378.]
Response 59:[Thank you for pointing this out. We agree with this comment.The change can be found on page---page 18, paragraph 1, line 691.
Regarding the issue of our use of qPCR in Figure 7 instead of further mining the RNA-seq data to explore the cold response of rye, there are several reasons for our choice of using qPCR. Firstly, qPCR serves as a verification tool for the results of RNA-seq. Taking into account the potential technical biases in RNA-seq, such as batch effects and inaccurate mapping, qPCR provides a reliable method to confirm the genes that are identified as differentially expressed in the RNA-seq analysis. Secondly, qPCR offers a higher level of quantitative precision. The precise quantification of gene expression changes is crucial for understanding the magnitude of the cold response. Finally, we have already mined the RNA-seq data to identify a subset of key genes that are likely to be involved in the cold response. Using qPCR enables us to focus on these specific genes, thus allowing for a more in-depth analysis of their expression patterns under cold stress.]